



# Where there is smoke there is mercury: Assessing boreal forest fire mercury emissions using aircraft and highlighting uncertainties associated with upscaling emissions estimates.

David S. McLagan[1,2], Geoff W. Stupple[1], Andrea Darlington[1], Katherine Hayden[1] and Alexandra
Steffen.[1,*]

[1]Air Quality Research Division (ARQD), Environment and Climate Change Canada, 4905 Dufferin
St, North York, ON M3H 5T4, Canada

[2]Dept. Environmental Geochemistry, Institute for Geoecology, Technical University of
Braunschweig, Langer Kamp 19c, 38106 Braunschweig, Germany

*Correspondence to*: Alexandra Steffen (Alexandra.steffen@canada.ca)

## ABSTRACT

Mercury (Hg) emitted from biomass burning is an important source of the contaminant to the
atmosphere and an integral component of the global Hg biogeochemical cycle. In 2018,
measurements of gaseous elemental Hg (GEM) were taken on-board a research aircraft along with a
series of co-emitted contaminants in the emissions plume of an 88 km$^2$ boreal forest wildfire on the
Garson Lake Plain (GLP) in NW Saskatchewan, Canada. A series of four flight tracks were made
perpendicular to the emissions plume at increasing distances from the fire each with 3 – 5 passes at
different altitudes at each downwind location. The maximum GEM concentration measured on the
flight was 2.88 ng m$^{-3}$, which represents a ≈2.4x increase in concentration above background. GEM
concentrations were significantly correlated with the co-emitted carbon species (CO, $CO_2$, and $CH_4$).
Emissions ratios (ERs) were calculated from measured GEM and carbon co-contaminants data. Using
the least uncertain of these ratios (GEM:CO), GEM concentrations were estimated at the higher 0.5
Hz time resolution of the CO measurements resulting in maximum GEM concentrations and
enhancements of 6.75 ng m$^{-3}$ and ≈5.6x, respectively. Extrapolating the estimated maximum 0.5 Hz
GEM concentration data from each downwind location back to source, 1 km and 1 m (from fire)
concentrations were predicted to be 12.9 and 29.9 ng m$^{-3}$, or enhancements of ≈11x and ≈25x,
respectively. ERs and emissions factors (EFs) derived from the measured data and literature values
were also used to calculate Hg emissions estimates on three spatial scales: (i) the GLP fires
themselves, (ii) all boreal forest biomass burning, and (iii) global biomass burning. The most robust
estimate was of the GLP fires (21 ± 10 kg of Hg) using calculated EFs that used minimal literature
derived data. Using a Top-down Emission Rate Retrieval Algorithm (TERRA) we were able to
determine a similar emission estimate of 22 ± 7 kg of Hg. The elevated uncertainties of the other


estimates and high variability between the different methods used in the calculations highlight concerns with some of the assumptions that have been used in calculating Hg biomass burning in the

literature. Among these problematic assumptions are variable ERs of contaminants based on vegetation type and fire intensity, differing atmospheric lifetimes of emitted contaminants, the use of only one co-contaminant in emissions estimate calculations, and the paucity of atmospheric Hg species concentration measurements in biomass burning plumes.

## 1    Introduction

A number of studies have provided evidence that mercury (Hg), a persistent, bioaccumulative, and toxic contaminant, is emitted during biomass burning (e.g. Friedli et al., 2003a; 2003b; Obrist et al., 2008; Chen et al., 2013). Emissions of Hg from biomass burning demonstrate one of the similarities between anthropogenically perturbed carbon and Hg biogeochemical cycles. The active pools of these elements in their respective biogeochemical cycles have been augmented by emissions from

anthropogenic activities such as mining and industry. Similar to carbon, plant biomass also represents a significant global sink of mercury emitted to the atmosphere. The major mechanism of Hg uptake to plants is the inspiration of gaseous elemental Hg (GEM; the dominant form of atmospheric Hg) via leaf stomata (Rea et al., 2001; Laacouri et al., 2013; Jiskra et al., 2015). While it was thought this process resulted in oxidation of the GEM taken up via leaf stomata leading to a relatively

unidirectional flux (Demers et al. 2013, Jiskra et al., 2015), a recent study using stable Hg isotopes suggests reduction and reemission of this internal leaf Hg (between 29 and 42 % of gross uptake based on the plant species studied) may occur (Yuan et al., 2018). The retained Hg in leaf matter associated with this uptake mechanism is eventually deposited to the ground in litterfall and either added to the pool of Hg in the soil or reemitted to the atmosphere during decomposition of the plant

material (St Louis et al., 2001; Demers et al., 2007; Demers et al., 2013).

Other possible uptake mechanisms of Hg to plant biomass have been considered and discussed in the literature. While gaseous oxidised Hg (GOM) and particulate bound Hg (PBM) can deposit to plant surfaces, in particular leaves, it has been suggested that this is not a stable sorptive process. Deposited Hg can be photo-reduced to GEM and reemitted to the atmosphere (Graydon et al., 2006; Mowat et

al., 2011; Demers et al., 2013) or washed off and deposited to soils by precipitation throughfall (Rea et al., 2000; 2001; Demers et al. 2007; 2013). It is also possible that plants can take up Hg from the soil via their roots (Godbold et al., 1988; St louis et al., 2001; Graydon et al., 2009). However, this process has been shown to contribute little to the accumulated Hg in biomass except in areas heavily contaminated with Hg (Lindberg et al., 1979; Graydon et al., 2009; Mowat et al., 2011).

The high volatility of elemental Hg (Ariya et al., 2015) and the conversion of oxidised forms of Hg to elemental Hg at temperatures generated in biomass burning (Biester and Scholz, 1996) results in





Hg stored in biomass being released to the atmosphere during biomass burning. Emissions of Hg from biomass burning are predominantly as GEM (Friedli et al., 2003a; Finley et al., 2009; De Simone et al., 2017). Emissions of GOM have not been detected from controlled or wildfire biomass burning plumes (Friedli et al., 2003a; Obrist et al., 2008; Finley et al. 2009; Chen et al., 2013). Nonetheless, GOM measurements have a lower temporal resolution and high inherent uncertainty (Finley et al., 2009; De Simone et al., 2017) and more measurements using a range of analysis methods are required to confirm this assessment. A key factor driving this uncertainty is the likelihood that GOM will partition to particles due to their elevated concentrations in biomass burning plumes (Obrist et al., 2008). While measurements of PBM are again uncertain due to differing methods, long sampling times, and other sampling artefacts (De Simone et al. 2017), emissions of PBM have been reported to contribute between 3.8 and 15 % to total atmospheric Hg (TAM) emissions in wildfires (Friedli et al., 2001; 2003a; 2003b; Finley et al., 2009; Chen et al., 2013) and from <1 – 48 % in controlled laboratory burns (Friedli et al., 2001; 2003a; Obrist et al., 2008). The proportion of PBM likely increases with increasing biomass moisture content (Obrist et al., 2008).

The proportion of stored Hg in biomass released to the atmosphere during combustion has been tested using a mass balance approach in controlled laboratory burns and is generally considered complete (>94 %), regardless of species (Friedli et al., 2001; Friedli et al., 2003a; Obrist et al., 2008). However, studies utilising controlled, laboratory burns consider only releases from burned living plant biomass and litterfall and are likely to underestimate actual emissions from wildfires that additionally include Hg released from underlying soils associated with soil heating (Friedli et al., 2003a). While large uncertainties remain as to the amount of Hg that is released from soils, DeBano (2000) reported that temperatures can reach 850 °C at the litter-soil interface in low organic content soils, but this rapidly decreases to approximately 150 °C at only 5 cm below the surface in dry soils. This suggests that Hg releases from soils are limited to the upper soil horizons (primarily the organic horizon; Engle et al., 2006; Biswas et al., 2008), where temperatures are likely to be sufficient (≥300 °C) to release at least a portion of, if not all, Hg complexed in soil organic matter (Biester and Scholz, 1996). Thus, Hg releases from soil are more likely to contribute an increased proportion of emissions in temperate and boreal forests, in which >90 % of total Hg in forest ecosystems can be contained in soil organic matter (Schwesig and Matzner, 2000; Friedli et al., 2007; Obrist, 2012).

While a number of studies have made atmospheric Hg measurements in biomass burning plumes, the majority of these studies have been based on measurements made at substantial distances from the fires themselves at either ground-based monitoring stations (Brunke et al., 2001; Sigler et al., 2003; Weiss-Penzias et al., 2007; Finley et al., 2009) or in aircraft (Artaxo et al., 2000; Ebinghaus et al., 2007; Slemr et al., 2018). From review of the literature, two studies were found that made aircraft-



based atmospheric Hg measurements directly in a biomass burning plume near-source (within 50 km of a fire). Friedli et al. measured GEM and PBM in wildfires in temperate forests in Northern Ontario, Canada (2003a) and Northern Washington State, USA (2003b) with GEM enhancements of up to ≈1.4 and 6 times background concentrations, respectively. Given carbon monoxide (CO)

concentrations are enhanced relative to atmospheric Hg in biomass burning compared to industrial plumes (Chatfield et al., 1998; Jaffe et al., 2005; Wang et al., 2015), these and other studies have used emissions ratios (ERs) of atmospheric Hg concentrations to co-located measurements of CO and/or carbon dioxide ($CO_2$) concentrations to identify biomass burning plumes. Additionally, ERs and/or emissions factors (EFs; unit mass of Hg released per unit mass of fuel combusted; g kg$^{-1}$) can be used

to make global biomass burning Hg emissions estimates using these more widely monitored carbon constituents emitted from biomass burning plumes as surrogates. Nonetheless, upscaling emissions using co-emitted surrogates requires some large assumptions (i.e. equivalent atmospheric residence times; ERs that do not vary by burning intensity) that can introduce considerable uncertainty to these estimates (Cofer III et al., 1998; Andreae and Merlet, 2001; Andreae, 2019).

In this study, we made aircraft-based measurements of GEM and co-emitted carbon gases in a plume from a Canadian boreal forest wildfire. It is our aim to assess the magnitude of GEM emissions from this fire, investigate ERs of GEM to CO, $CO_2$, methane ($CH_4$), and non-methane hydrocarbons (NMHCs), each enhanced in biomass burning plumes, and to estimate total boreal/temperate forest and global emissions for Hg from biomass burning based on these data using four different upscaling

methods. We also assess the validity of upscaling these emissions estimates, highlighting the uncertainties associated with these calculations.

## 2 Methods

### 2.1 Site and flight descriptions:

The forest fire was situated at approximately 56.45 °N and 109.75 °W (425 – 450 m a.s.l.) on the

Garson Lake Plain (GLP) between Garson Lake and Lac La Loche in Northern Saskatchewan, ≈520 km NNW of Saskatoon, Canada (≈400 km NNE of Edmonton; Figure 1). The fire was ignited by a lightning strike and burned from the 23$^{rd}$ to the 26$^{th}$ of June 2018, burning a total area of ≈88.0 km$^2$ (a 10 % uncertainty is assumed with this estimate). The total burned area was calculated using satellite imagery (NASA, 2020) and ArcGIS (ESRI) and can be found in the supplemental information (Fig.

S1.1). The area burned is part of Canada's Boreal Plains biome and is a mixed northern boreal forest likely dominated by black spruce (Picea mariana), tamarack (American larch; Larix laricina), trembling aspen (Populus tremuloides), and jack pine (Pinus banksiana) (Korejbo, 2011; Nesdoly, 2017). Other tree species such as white spruce (Picea glauca), balsam poplar (Populus balsamifera),



balsam fir (Abies balsamea), and paper birch (Betula papyrifera) may also have been present in the

forest stands burned in this fire (Korejbo, 2011; Nesdoly, 2017). Although this fire occurred close to the Alberta Oil Sands' facilities (≈100 km ESE of Fort McMurray; main urban centre of the oil sands operations), winds during this flight were relatively stable south-easterlies ($136 \pm 10$ °). As such, all segments of the flight were upwind of all facilities of the Alberta Oil Sands and the data should not be influenced by any emissions of Hg from these facilities.

Measurements of GEM, CO, $CO_2$, $CH_4$, and NMHCs were made on board the National Research Council's (NRC) Convair-580 research aircraft in the plume of the GLP fire on June 25[th], 2018. The monitoring component of the flight occurred between 15:00 and 18:58 GMT (09:00 and 12:58 in local mountain daylight time in Alberta). Analysis of the fire plumes and thermal anomalies of the MODIS satellites imagery confirms the fire peaked on June 25[th], 2018 (NASA, 2020). The flight

comprised of a number of transects at different altitudes that passed through the plume perpendicular to the plume direction to create a virtual screen. Four screens were completed at successive distances downwind of the fire source (Fig. 2). The middle of the plume was calculated to be approximately 5 – 20, 30 – 45, 55 – 70, and 85 – 100 km from the burning fires for screens 1, 2, 3, and 4, respectively. Difficulties in constraining these distances relate to the multiple fires burning on the day of the

monitoring flight (Figure 2). The middle of this range was used in calculations based on these data. The number of transects for each screen was 5, 4, 4, and 3 for screens 1 – 4, respectively. A vertical spiral was flown during each screen to determine the vertical extent and structure of the plume and the height of the mixed layer. The mean wind speeds and temperatures measured on the aircraft during the flight were $7.9 \pm 2.4$ m s[-1] and $20.4 \pm 4.1$ °C, respectively. The closest weather station to these

fires was Lac La Loche weather station (≈23 km east of the fires on the eastern side of Lac La Loche; 56.45 °N, 109.40 °W) and the mean hourly ground wind speed, temperature, and relative humidity measured during the flight were $4.1 \pm 2.4$ m s[-1], $25.8 \pm 2.0$ °C, and $58.0 \pm 12.0$ %, respectively (ECCC, 2019). Daily average wind speed, temperature, relative humidity, precipitation, and fire danger determinants for the week preceding the flight at this station are provided in Section S2.

## 2.2 Gaseous elemental mercury measurements:

The NRC's Convair 580 research aircraft was fitted with a Tekran 2537X instrument (Tekran Instruments Corporation) for measuring GEM. The system sampled GEM and a detailed discussion of the determination of GEM as the sampled analyte is given in the supplementary information (Section S3). General details of this instrument can be found in Cole et al. (2014). The instrument

was setup for in-flight use to decrease sample time and reduce uncertainties that can arise during aircraft deployments due to pressure changes (e.g. Slemr et al., 2018) and specific details pertaining





to this study are as follows. A shortened analytical cycle was developed and successfully tested in the lab (no loss of instrument accuracy and precision) that used a shorter (25 sec), but higher flush rate (0.2 L min⁻¹) along with shortened cartridge heat times (15 sec) and cooling time (30 sec). This

shortened analytical timing allowed for a shorter sample time of 2 min with a system flow rate of 1.5 L min⁻¹ to give a measured sample volume of 3 L. To avoid changes in pressure affecting the cell flow, a pressure controller was used on the cell vent and maintained at a constant pressure slightly above ambient ground pressure. Ambient air was drawn in through a rear-facing inlet (to prevent particles entering the inlet) mounted on the roof of the aircraft. This inlet incorporated a by-pass

system that flooded the inlet with "zero" air generated by a series of three activated-carbon filters into the instrument during take-off and landing to prevent contamination. The inlet line was 5.44 m in length from the inlet to the instrument and made from PTFE with an inside diameter of 2.5mm. Along with sampling lines for other gaseous species this was heated to 50°C for the first 4.5 m to prevent moisture from condensing within the sampling line. The remaining unheated sampling line

incorporated a soda-lime trap fitted at each end with cleaned quartz wool to remove water vapour and acidic gases, as well the standard Tekran 2537 series filter pack containing a 0.25 µm Teflon filter.

The system was running for a period of >72 hr both before and after the flight to ensure the system was at its optimal stability. During this time, the system sampled Hg mercury free air generated by a Tekran 1100 zero air generator (Tekran Instruments Corporation). Approximately two-hours before

take-off, a series of three 55.7 pg Hg additions from the internal permeation unit of the system were made on each of the two gold amalgamation traps (additions every third sample). The additions equated to a GEM concentration of 18.57 ng m⁻³ in a 3L sample. This process was again repeated after the flight. These additions function in the same way as the normal Tekran 2537X calibrations and were used to calibrate the system for the flight. The measured concentration for any given

sample was adjusted using a linear adjustment based on the mean of the additions for each trap before and after the flight proportional to when the sample was taken within the flight according to Equation 1:

$$C_i = Z_i / [Y_i - (Y_i - X_i) * \left(\frac{A}{B}\right)] \tag{1}$$

Where, $C_i$ is the reported GEM concentration measured on trap $i$; $Z_i$ is the instrument signal (area counts) for a sample measured on trap $i$;

counts) for a sample measured on trap $i$; $Y_i$ is the mean calibration factor (instrument signal for the addition divided by the expected concentration) for the additions made on trap $i$ before the flight; $X_i$ is the mean calibration factor for the additions made on trap $i$ after the flight; $A$ is the number of each specific measurement ($A$=1 for the first measurement of the flight); $B$ is the total number of measurements taken during the flight; $i$ has values of 1 or 2 according to which gold trap the sample





was amalgamated on within the Tekran 2537X. This calibration method was used to account for any instrumental drift that may have occurred during this unique in-flight deployment. The additions before this particular flight were 7.3 % higher than after the flight; hence the calibration method applied corrected for this drift. Before and after the campaign the internal permeation source was verified using manually injected $Hg^0$ from a temperature-controlled Hg vapour source at saturation

vapour pressure. Recovery from these injections were $98.7 \pm 0.7$ %.

Due to power and space constraints, no atmospheric Hg speciation measurements could be made on this flight. All references to measurements made by Tekran 2537 series instruments from other studies: either GEM or total gaseous Hg (TGM = GEM + GOM) will be referred to as GEM for clarity and consistency purposes. As previously described, GOM has not been measured to be elevated above

background in wildfire biomass burning emissions (Friedli et al., 2003a; Obrist et al., 2008; Finley et al. 2009; Chen et al., 2013). Thus, any differences between GEM measurements from this study and TGM measurements from other studies based on those studies potentially sampling some GOM are likely to contribute only a minor uncertainty to any data comparisons. All GEM concentrations from this study are reported on a mass-per-volume basis with mixing ratios also reported in parentheses.

Mass-per-volume to mixing ratio conversion calculations used standard temperature and pressure as the mass flow controller of the Tekran 2537X instrument has already adjusted the mass-per-volume concentrations for the actual temperature and pressure during each measurement cycle.

### 2.3   Measurements of other air pollutants:

CO, $CO_2$, and CH4 were measured with a Picarro G2401-m instrument based on cavity ring down

spectroscopy. Calibrations were performed at the beginning and end of each flight using calibration gas mixtures at two different mixing ratios. The NMHCs were measured with a difference method using two Picarro G2401-m instruments. One instrument sampled through a heated catalyst that converted all the atmospheric C species, including $CO_2$, CO, $CH_4$ and NMHC to $CO_2$ and the second instrument measured $CO_2$, CO and $CH_4$ in ambient air (not through the catalyst) and these mixing

ratios were used to subtract from the first instrument to obtain a measure of NMHCs.  This method was adapted from Stockwell et al. (2018). To allow data comparisons between GEM and these other species that are measured at greater frequency, all CO, $CO_2$, CH4, and NMHC data were averaged to the same 2-minute sampling resolution of the Tekran 2537X instrument.

### 2.4   Emissions ratios, emissions factors, and emissions estimates
calculations:

Background concentrations of the contaminants is required in certain components of the emissions estimate calculations. For GEM this was determined to be $1.18 \pm 0.02$ ng m$^{-3}$ ($1.31 \pm 0.02$ x $10^{-7}$ ppm)



during this flight based on the mean measurements made outside the biomass burning plume. The equivalent background concentration data for the same sampling period for CO, $CO_2$, $CH_4$, and

NMHCs were $0.134 \pm 0.022$ ppm, $405.2 \pm 1.0$ ppm, $1.906 \pm 0.005$ ppm, and $0.107 \pm 0.091$ ppm, respectively. All emissions ratios (ERs), emissions factors (EFs) and emissions estimates were based on GEM concentrations that were enhanced by >125 % of the background GEM concentration; data below this fraction were too variable and uncertain, particularly for the $CO_2$ enhancements due to the more elevated and variable background concentration of $CO_2$ (Yokelson et al., 2013; Andreae, 2019).

In total there were 24 GEM concentration measurements enhanced by >125 % of background.

The ER is the slope of the regression of a target species ($X$) and a reference species ($Y$), preferably both enhanced in an emissions plume according to Equation 2 (Jaffe et al., 2005):

$$X = ER_{XY} * Y \tag{2}$$

Both the $\Delta X : \Delta Y$ (excess mixing ratios, adjusted for background) and $X : Y$ (measured mixing ratios)

ratios have been used in previous studies. However, regressions of both relationships generate the same slope. Here we will use the unitless ERs based on the mixing ratios of GEM to CO, $CO_2$, $CH_4$, and NMHC unadjusted for background concentrations in order to display the original data.

It is also possible to calculate ERs using an integration method (Urbanski 2013). ERs using this method for GEM:CO, GEM:$CO_2$, GEM:$CH_4$, and GEM:NMHCs were within 10 % of the regression

method – consistent with variability in the literature (Urbanski 2013). The ERs determined using the regression method (Equation 2) are used in this study.

EFs (unit mass of Hg released per unit mass of fuel combusted; g $kg^{-1}$) are also an important component required to estimate Hg emissions from biomass burning. These can either be estimated by adjusting the measured ERs relative to the more widely known EFs of reference species and each

compound's molecular weight ($MW$; Equation 3; Andreae and Merlet, 2001; Andreae, 2019):

$$EF_X = ER_{XY} \frac{MW_X}{MW_Y} EF_Y \tag{3}$$

or using the measured data based on Equation 4 (Andreae and Merlet, 2001):

$$EF_X = \frac{\Delta X * MW_X}{[(\Delta CO + \Delta CO_2 + \Delta CH_4 + \Delta NMHC) * MW_C]} * C_{biomass} * 1000 \tag{4}$$

$MW_C$ is the molecular weight of carbon, and $C_{biomass}$ is the fraction of carbon in biomass. The latter

has been assumed as 0.45 in Hg biomass burning emissions estimates in boreal/temperate forests, but no uncertainty in this parameter is given (Friedli et al., 2003b). Thurner et al. (2014) report higher carbon contents in boreal needleleaf forests (the majority of species in the burned stands of the GLP



are needleleaf) of 0.508 with a "negligible" uncertainty. We will use this value in our emissions estimate calculations with an assumed 5 % uncertainty (0.508 ± 0.025) for error propagation purposes.

It is important that we consider that the ERs calculated from the GEM concentration data do not include any PBM fraction. All our emissions estimates include TAM scenarios of 0, 3.8, 15, and 30 % PBM with the remainder being our measured GEM concentrations (no GOM contribution) to cover the range of uncertainty associated with the unmeasured and otherwise uncertain PBM fraction. The 0 % PBM scenario produces GEM emissions estimates based directly on our measured GEM
concentration and represents the lowest data uncertainty; these are the data predominantly discussed in this study. The 3.8 % PBM scenario equates to the measured fraction from Friedli et al. (2003b), which represents the most relevant near-source, aircraft-based monitoring of Hg in a wildfire plume and allows direct data comparison between this and their study. The 15, and 30 % are also assessed for model sensitivity purposes and are the assumed fraction and suggested upper limit of the PBM
fraction in De Simone et al. (2017), respectively. Adjustments for PBM were achieved by dividing the GEM concentration data by one minus the assumed fraction of PBM, then recalculating the regressions between GEM and the other primary pollutants.

There are a number of methods that can be used to estimate Hg emissions from this wildfire and potentially upscale this to estimate emissions of Hg for regional or global boreal/temperate forests
and even global emissions from all biomass burning sources based on the calculated ERs and EFs. To stay within the scope of our study we will constrain our emissions estimates to four simpler methods and leave more complex emissions modelling for future studies. The mean burned areas used for upscaling emissions to all boreal forests and for total global biomass burning are $78 \pm 50$ x $10^4$ and $3.49 \pm 0.24$ x $10^6$ $km^2$ $yr^{-1}$, respectively, and were derived using the GFEDv4 model
(Randerson et al., 2018) and the data were taken from Giglio et al. (2013) for 1995 - 2011.

Emissions estimate method 1 (EEM1) is the most basic method and simply takes the estimated global emissions of the three more widely monitored carbon gases described previously (CO, $CO_2$, and $CH_4$) and adjusts these emissions estimates according to the measured ERs in our study. The estimated CO, $CO_2$, and $CH_4$ emissions taken from the literature are given in Section S5 (Table S5.1; Jiang et al.,
2017; Shi and Matsunaga, 2017; Worden et al., 2017). This method cannot produce an estimate for the GLP Fires monitored in this study.

Emissions estimate method 2 (EEM2) uses Hg EFs derived from the reference pollutant EFs (see Section S6 and Andreae, 2019 for the EF values used) based on Equation 3. The emission estimate ($Q_x$) is then calculated according to Equation 5:

$Q_X = A * B * F * EF_X$ (5)





Where $A$ is the total burned area, $B$ is the fuel load and is assumed to be $2.35 \pm 0.99$ kg m$^{-2}$ (mean fuel load burned in all fires in Canada's Boreal Plains, 1959 – 1999; Amiro et al., 2002), $F$ is the fraction of Hg released and is 1.0 as it is assumed all Hg is released during the fire (with an assumed 0.05 error term to this value). Similar to EEM1, EEM2 will have Hg emissions estimates based on

CO, $CO_2$, and $CH_4$ as reference compounds.

Emissions estimate method 3 (EEM3) also uses Equation 5 and is the same as EEM2 except that the EFs are calculated from the measured data according to Equation 4. The calculated EFs used in EEM2 and EEM3 are listed in Section S6 (Table S6.1).

The final method uses a Top-down Emission Rate Retrieval Algorithm (TERRA) and has been

designed to generate emissions data specific to the aircraft measurements that were made in this study (Gordon et al., 2015). As such, it is used to evaluate the emissions estimates for the GLP fires and considered separately to the discussion regarding the assessment of upscaling emissions estimates. TERRA estimates emissions transfer rates (kg hr$^{-1}$) through boxes or screens from aircraft measurements using the divergence theorem. Pollutant and wind data are mapped to a virtual screen

and concentration data interpolated using a simple kriging function. In this study we apply TERRA to the stacked horizontal legs of the flight track on the first screen downwind of the fire. Concentrations of Hg are extrapolated below the lowest flight altitude using a linear least-squares fit at each horizontal grid square below the lowest flight track in the plume area. Extrapolation below the flight path has been shown to be the main source of uncertainty in TERRA. Alternate

extrapolations were tested including assuming a well-mixed layer below the flight path and assuming a background concentration at the surface and a linearly decreasing concentrations between the lowest flight track and the surface. There was less than 5 % difference in the resulting emission rates between these three methods of extrapolating data to the surface. More detailed uncertainty estimations for TERRA is contained in Gordon et al., (2015) and Liggio et al., (2016). To produce an emissions

estimate for the whole fire using TERRA, the emissions transfer rate was upscaled by two methods. (i) Assuming constant emissions transfer rate across the whole burning area. (ii) Assuming this was the mean emissions transfer rate ($QR_x$) for the day of the flight (25$^{th}$ of June), and adjusting emissions from other days and nights by multiplying the emissions rate by the ratio of MODIS satellite fire hot spots observed on those days ($n_{iD}$) or nights ($n_{iN}$) compared to the number of fire hotspots in the day

of June 25$^{th}$ ($n_{25}$) (Eq. 6). Eq. 6 assumes 6-hour night and 18-hour day of this high latitude location in mid-summer.

$$Q_X = (QR_X * 18) + (QR_X * [{}^{n_{1D}}/_{n_{25}}] * 18) + (QR_X * [{}^{n_{1N}}/_{n_{25}}] * 6)) + \ldots + (QR_X * [{}^{n_{iD}}/_{n_{25}}] * 18) +$$
$$(QR_X * [{}^{n_{iN}}/_{n_{25}}] * 6)) \tag{6}$$



We list all data taken from literature with one extra significant digit (where possible) to reduce
rounding uncertainty in these calculations. Overall uncertainties of emissions estimates were
calculated using error propagation according to Eq. 7:

$$\sigma_T = \left[ \sqrt{\left(\frac{\sigma_a}{a}\right)^2 + \left(\frac{\sigma_b}{b}\right)^2 + \cdots + \left(\frac{\sigma_i}{i}\right)^2} \right] * T \qquad (7)$$

Where, $a$, $b$, … , $i$, and $T$ are the estimates for each variable and the total, respectively; and
$\sigma_a$, $\sigma_b$, … , $\sigma_i$, and $\sigma_T$ are the standard deviations or error estimates for each variable and the total,
respectively. All statistical testing and calculations were performed using OriginPro 2018
(OriginLab).

## 3   Results and Discussion

### 3.1   Elevated gaseous elemental mercury concentrations:

Measurements taken on-board the NRC's Convair 580 research aircraft during the GLP fires showed
GEM concentrations elevated above background in all four of the screens of the flight on June 25[th]
2018 (Figure 2 and Figure 3). The plume was divided into a north and south plume, whose
approximate paths are described by the yellow and orange-dotted lines in Figure 2(a), respectively.
This was likely caused by shifting overnight winds that changed plume trajectory. While, there is the
possibility of the NP being derived from an additional fire source not detected by satellite, analysis
of satellite imagery in the days before and after the flight provide no evidence of this (no additional
source plumes or burned areas near GLP). Considering all data from the whole flight, the GEM
concentration was highly correlated with other primary pollutants emitted throughout this flight: CO
($R^2 = 0.983$; $p = 1$ x $10^{-105}$), $CO_2$ ($R^2 = 0.801$; $p = 3$ x $10^{-43}$), $CH_4$ ($R^2 = 0.736$; $p = 6$ x $10^{-36}$), and
NMHCs ($R^2 = 0.820$; $p = 8$ x $10^{-46}$) confirming these fires as a primary source of GEM to the
atmosphere (Fig. 3(a)). The maximum GEM concentration was measured in the south plume at 2.88
ng m$^{-3}$ (3.22 x $10^{-7}$ ppm) and occurred during the second transect of Screen 1 at ≈280 m above the
ground (710 m a.s.l.). This represents up to a 2.4x increase in GEM concentrations inside the biomass
burning plume during Screen 1. The maximum GEM concentrations measured for the subsequent
screens were always in the south plume and were 2.70, 2.36, and 1.73 ng m$^{-3}$ (3.19 x $10^{-7}$, 2.63 x $10^{-7}$
$^{7}$, and 1.93 x $10^{-7}$ ppm), representing enhancements of 2.3, 2.0, and 1.5x above background for
Screens 2 – 4, respectively.

The two other studies examining GEM concentrations in near-source, wildfire plumes using aircraft
measured enhancements of ≈1.4 (Friedli et al., 2003a) and ≈6 (Friedli et al., 2003b) times background,
placing the maximum enhancement observed in our study in the middle of those values. Both previous
studies appear to have sampled the emissions plumes closer than Screen 1 of our flight. Considering





we are likely to have measured the maximum concentration (or very close to it) in the plumes in our study due to the multiple passes at different altitudes in each screen, it is apparent there is variability in the enhancement of GEM concentrations and emissions from the wildfire plumes of these different studies. This may be attributable to the extent of area burning during the sampling period, intensity

(flaming or smouldering; potential change in PBM fraction) of the fire during the monitoring period, and/or variability in the concentration of Hg in the biomass of the different tree species being burned. The wildfire in Northern Ontario measured by Friedli et al. (2003a) was much smaller with a total burned area of only 1.7 km² and this difference was likely the key driver of the much lower enhancements observed. The Washington State wildfires monitored by Friedli et al. (2003b) burned

an area of 220 km² (data from Biwas et al., 2008), 2.5x greater burned than the GLP fires. Post take-off and between plume GEM concentrations in the earlier parts of their flight were considerably elevated (between ≈1.8 and 4 ng m⁻³) compared to post plume GEM concentrations at the end of the flight (≈1.2 ng m⁻³) despite the authors reporting the fire intensity increased throughout duration of the flight (Friedli et al., 2003b). The authors could not explain the background changes (Friedli et al.,

2003b), but it could be an artefact due to some instrument system contamination at the start of the flight, which may have increased the maximum measured concentrations in that study. Measurements collected from the ground-based Cape Point monitoring station in South Africa are the only other near source measurements reported from a wildfire emissions plume (23 km NNW of the site). This fire burned a very similar area to the GLP fires (≈ 90 km²) and GEM enhancements were ≈1.45x

background (Brunke et al., 2001).

### 3.2 Emissions Ratios:

ERs are based on the assumptions that there is no chemical (reaction) or depositional losses of one or both of the measured contaminants and that there is equivalent and constant dilution (Jaffe et al., 2005; Yokelson et al., 2013). This is a valid assumption for measurements taken in biomass burning

emissions plumes near source such as those of our study as negligible atmospheric reactions or deposition will occur for any of the considered species (GEM, CO, $CO_2$, $CH_4$, or NMHCs). The ER for GEM:CO based on the data with GEM enhancements of >125 % of background for the GLP fires displayed in Figure 3(b) and

(which equates to 0.83 ± 0.03 ng m⁻³ ppm⁻¹ using mass-per-volume concentration for GEM) had the

strongest fit of the four carbon contaminants examined with an $R^2$ value of 0.979. GEM:CO ERs are also the most commonly used in the literature to examine Hg emissions from biomass burning. Wang et al. (2015) summarised the use of GEM:CO ratios from all biomass burning studies and showed a range from 6.7 ± 0.4 x 10⁻⁸ taken by near-source aircraft measurements in the Washington State fires ($R^2$ = 0.86; Friedli et al., 2003b) up to 2.4 ± 1.0 x 10⁻⁷ using a commercial aircraft at an unknown





distance from non-specific fires ($R^2$ = 0.54; Ebinghaus et al., 2007). This places the GEM:CO ER determined in our study near the lower end of this range, but 1.3x higher than the other near-source aircraft measurements taken in the large fires in Washington State. The GEM:CO ER of the other near source aircraft-based study (Northern Ontario fire) was 2.2x that of our value, suggesting enhanced GEM emissions in the small Northern Ontario fire. Our data has the lowest uncertainty of

any of the previous studies (Wang et al., 2015), which gives us confidence in our data and this GEM:CO ER.

As previously mentioned, many of the studies that have addressed Hg in biomass burning are not near-source measurements, but rather long-range transport of pollutants from the fire sources to distant receptor sites. For any assessment of ERs and emissions estimates to be valid, the ER of the

two emitted species must remain constant even after long-range transport of both contaminants. While CO has been suggested to have a lifetime of several months (Khalil and Rasmussen, 1984; Yurganov et al., 2005; Turnbull et al., 2006), it can be significantly reduced to as little as 10 days in summer over continental landmasses (Holloway et al., 2000; Yurganov et al., 2004). Although GEM can be readily oxidised under very specific atmospheric conditions (coastal sites in polar spring; Steffen et

al., 2002; conditions not met in the current study), the lifetime of GEM is generally accepted to be ≈4 – 12 months (Holmes et al., 2010; Horowitz et al., 2017; Saiz-Lopez et al., 2018). This difference in lifetime suggests that CO could be more readily lost from the atmosphere than GEM. Since the majority of biomass burning occurs in summer months, such differences undermine the assumption that the ER will be conserved during long-range transport. This becomes progressively more

problematic as the distance between source and receptor sites increases. Consequently, the majority of studies that have estimated GEM:CO ER at large distances from the biomass burning source are likely overestimating GEM:CO ERs and is the likely explanation for the higher ERs reported in such studies (Wang et al., 2015). Potential differences in atmospheric lifetimes between these two primary biomass burning contaminants has not been critically discussed previously in literature on Hg

emissions from biomass burning.

Differences in lifetimes of GEM and CO are therefore not the major factor behind the differences between the GEM:CO relationship between the GLP fire and Cape Point wildfires in South Africa in which the ground-based monitoring station was only 23km from the burning source (Brunke et al., 2001). GEM:$CO_2$ ER has also been addressed in other studies and the GEM:$CO_2$ ER calculated in

the GLP fires is slightly lower than the ratio measured by Brunke et al. (2001) in South Africa. Brunke et al. (2001) also derived a CO:$CO_2$ ER of for their fire, which is ≈2x lower than the CO:$CO_2$ ratio measured in our study. Given the GEM:CO ER measured by Brunke et al. (2001) was 2.3x higher than in our study, it is evident that the CO emissions are either depleted in the South African fire or

enhanced in the GLP Fire (this study) in relation to both GEM and $CO_2$. Interestingly, $CO:CO_2$ ERs

430 from both the South African (see Hao et al., 1996; Koppmann et al., 1997) and the GLP (see Friedli et al., 2003a; Simpson et al., 2011) wildfires agree well with the corresponding ratio measured in plumes of fires that burned similar vegetation in their respective regions. Emissions of CO can vary relative to other emitted contaminants by fuel type (vegetation), burning stage or intensity, and even the period of the burning season (Cofer III et al., 1998; Korontzi et al., 2003). The GLP fires were

435 relatively low intensity, ground-based, smouldering fires, which causes increased emissions of CO – an incomplete combustion by-product (Lapina et al., 2008). Variability in the proportion of CO released from biomass burning is likely a major factor driving the variability of GEM:CO ERs in the literature. Nevertheless, it must also be noted that using $CO_2$ as a reference compound in ERs can also be problematic as the fraction of the $CO_2$ enhancement relative to background is less than other

440 contaminants and $CO_2$ background concentrations are more variable (Yokelson et al., 2013; Andreae, 2019). This explains the greater scatter of data observed for the GEM – $CO_2$ regression in the GLP fires (R2 = 0.750; Figure 3).There may be other primary pollutants that can be used to better comprehend Hg emissions from biomass burning. $CH_4$ is enhanced in biomass burning plumes, has a long atmospheric lifetime (≈9 yrs; Daniel and Solomon, 1998; Montzka et al., 2011), and it varies

445 less than CO based on vegetation type and fire intensity (Cofer III et al., 1998; Korontzi et al., 2003). Nonetheless, the GEM:$CH_4$ ER measured in the GLP fire carries a poorer fit (greater uncertainty; $R^2$ = 0.671) than both the GEM:CO and GEM:$CO_2$ ratios (Figure 3(b); Table 1). Hence, on its own does not represent an improved single reference compound in the estimation of Hg emissions. The fit of the GEM:NMHC ER ($R^2$ = 0.814) was better (lower uncertainty than both GEM:$CH_4$ and GEM:$CO_2$

450 ERs) and indeed contributed more to the fraction of carbon released from the GLP fires (mean fraction: 9.2 % of the considered elevated data) than $CH_4$ (mean fraction: 1.3 %). However, this ratio is unlikely to be efficacious at receptor sites distant from burning sources due to the variability in atmospheric lifetimes of the many compounds that make up NMHCs. This study represents the first time GEM:$CH_4$ or GEM:NMHCs ERs have been examined in the literature.

455 Given the strong linear fit of the regression between GEM and CO mixing ratios (higher $R^2$ and lower *p*-value; Figure 3) and the greater proportional enhancement of CO, the GEM:CO ER was used to estimate GEM concentrations at the higher time resolution of the CO data (0.5 Hz). The maximum estimated GEM concentration derived was 6.75 ng m$^{-3}$, (7.54 x 10$^{-7}$ ppm), which represents a 5.6x enhancement compared to the background GEM concentration (Figure 4). This data was also used to

460 generate the three-dimensional GEM concentration flight path in Fig 2(a).

McLagan et al. (2018; 2019) used power relationships between GEM concentrations and distance from source to estimate the concentrations at (1 m from) point sources. In these studies, passive





samplers were used to measure GEM concentrations, which involved longer deployments and provided time-averaged concentrations that were unable to ensure measurements were always

downwind of source. Concentrations decreased more rapidly with distance from source than what was observed in the current study (McLagan et al., 2018; 2019). Based on the estimated 0.5 Hz GEM concentration data from the GLP fires, a logarithmic relationship ($R^2 = 0.998$; Figure 4(b)) was used to project GEM concentrations at the wildfire source as it produced a stronger fit than a power relationship ($R^2 = 0.976$). The estimated concentrations were 12.9 (1.44 x $10^{-6}$ ppm) and 29.9 ng m$^{-3}$

(3.33 x $10^{-6}$ ppm) at 1 km and 1m from the fires, respectively. This would represent 11x and 25x GEM enhancements above background, respectively. While these modelled GEM concentration estimates come with expectedly high uncertainty, they elicit otherwise unattainable information on the GEM concentrations at the active source of these wildfires. Contributing factors to this uncertainty include uncertainties in ER calculation, extrapolation of the logarithmic concentration – distance

relationship, uncertainty of exact distances the measurements were made from the fires (wildfires are not a single point source), and variable wind speeds during the sampling period.

### 3.3   Mercury emissions estimates:

The emissions estimates for Hg from biomass burning using EEM1, EEM2, and EEM3 are listed in Table 2. Estimates of GEM emissions from the GLP fires ranged from 12 ± 8 kg using EEM2 and

CH$_4$ as a reference compound to 21 ± 10 kg using EEM3 (Table 2). Differences up to 1.8x between the GEM emissions estimates for the GLP fires using these two methods demonstrates the increased uncertainty of emissions estimates that arises when assuming literature based ERs data (EEM2) in these calculations. EEM3 is the only method applied here that does not use literature derived EFs or ERs from reference contaminants to determine Hg emissions. The only assumed values from the

literature applied in EEM3 are the fraction of carbon in the biomass burned that has a low inherent uncertainty (because it has been extensively assessed due to the importance of carbon in biomass and carbon emissions from biomass burning) and the fuel load of the area burned. The latter value does have considerable uncertainty (our value for Canadian Boreal Plains forests has an uncertainty of 42 %) as it is exceedingly difficult to predict where fires will occur and assay the fuel load of the exact

burned stands pre-emptively. Nonetheless, fuel load of area burned is an assumption that must be made in all estimates. Thus, we deem the Hg emissions estimates for the GLP fires to be the most appropriate method contextualised by its 52 % propagated uncertainty, a large factor of which is derived from the assumed fuel load.

Friedli et al. (2003a; 2003b) estimated Hg emissions using EEM3, albeit with some different

assumptions. While we cannot directly compare Hg emissions from these fires to our emissions estimate of the GLP fires due to differences in burned areas, the aforementioned studies did produce





emissions estimates for boreal forests of 59.5 Mg yr$^{-1}$ (no uncertainty given; Friedli et al., 2003a) and

22 Mg yr$^{-1}$ (no uncertainty given; Friedli et al., 2003b). The estimate made by Friedli et al. (2003b),

which includes their measured 3.8 % PBM fraction, is similar to our EEM3 estimate for Boreal

Forests when we add the same assumed 3.8 % PBM fraction to our GEM data (19 ± 15 Mg yr$^{-1}$). The

higher emissions estimate made from the small Northern Ontario fire (Fiedli et al., 2003a) is likely

related to the previously discussed GEM enhancement (relative to CO and CO$_2$) of that particular

fire. The EFs of all three studies (Friedli et al., 2003a; 2003b and our study) are also similar (Table

1).  However, an important difference between these studies and the GLP fires is the assumption by

Friedli et al. (2003a; 2003b) of a fixed ratio of carbon species in the emissions plume of 10:90:0:0

(CO:CO$_2$:CH$_4$:NMHC). In contrast, the mean ratio of carbon species in the elevated data (>125 % of

GEM background) in the GLP fires was 13.0:76.5:1.3:9.2 (± 3.4:6.1:0.4:3.5; CO:CO$_2$:CH$_4$:NMHC),

respectively. If we assume the same 10:90:0:0 ratio of carbon contaminant emissions (derived from

our measured CO concentrations only), the 3.8 % PBM EF becomes 80 ± 9 µg kg$^{-1}$ for the GLP fires

(see Section S6, Table S6.1). This 10:90:0:0 EF is 1.4x lower than the EFs in either the Northern

Ontario or Washington State fires., which is similar to the difference in ERs between the GLP 1.4x

higher) and the Washington State Fires. As Friedli et al. (2003b) report, the EEM3 calculation is

highly sensitive to the ratio of carbon species emitted; changes in this ratio, which can be indicative

of variable burn intensity (Cofer III et al., 1998), can have an exponential effect on the emissions

estimate. This highlights the increased uncertainty associated with the use of a single reference

compound and assumed ratios of carbon species emitted in deriving Hg emissions estimates.

Furthermore, the elevated carbon fraction made up by NMHCs in the GLP fires brings into question

the assumption that CO, CO$_2$, and CH$_4$ make up >95 % of carbon emissions (Fiedli et al, 2003b;

Urbanski, 2013), particularly for smouldering fires such as these that can lead to an increased

proportion of NMHC emissions (Urbanski, 2013).  Recent studies with updated NMHC methods

(such as the system used in this study) confirm that NMHCs have been "severely" underestimated in

earlier literature on biomass burning emissions (Andreae, 2019).

Similar to the studies by Friedli et al. (2003a; 2003b), the EF derived from the GLP fires is higher

than those measured from laboratory studies (Friedli et al., 2001; 2003a; Obrist et al., 2008). As

Friedli et al. (2003a) suggest, this is likely to be caused by the additional Hg emissions from upper

soil layers in the wildfires. Soil components have generally not been included in controlled laboratory

burns addressing Hg biomass burning emissions.

The assumptions of fuel load and biomass carbon fraction are derived from data for boreal forests,

and similarly our measurements are of a boreal forest fire. Thus, we suggest our EEM3 estimates to

be the most relevant to Hg emissions from global boreal forests. Even though the EEM1 and EEM2





estimates take data from the literature based on boreal forests, they rely on externally sourced emissions-related data based on an uncertain single reference compound. All the boreal forest emission estimates do; however, have the highest uncertainty of the three emissions scales. This elevated uncertainty is largely associated with the large interannual variability in burned area of
boreal forests in North America and Asia (Fraser et al., 2018). The high variability of this estimate must be incorporated into any boreal forest emissions estimate.

Highly constrained global Hg emissions estimates represent an end-goal of research into emissions of Hg from biomass burning. Nevertheless, global scale emissions introduce a new set of challenges that are not present when assessing emissions from a single fire or single forest type: chiefly,
differences in vegetation type (biome) and meteorology and the associated variability in fire behaviour caused by these differences (Kilgore 1981; Hély et al., 2001). As stated, the variables used in the EEM3 calculation are tailored to boreal forests; hence, the applicability of this method becomes problematic for global scale emissions estimates. EEM1 and EEM2 use the measured ER from the GLP boreal forest fires, and hence introduce similar concerns associated with up-scaling data drawn
from a single biome. The range of estimated Hg emissions made using the three methods are highly variable and differ by up to a factor of 6.1 (Table 2). While coefficient of variation (values in parenthesis in Table 2) for the global estimates are lower than for the GLP fires or boreal forest fire emissions estimates using single reference compounds (EEM1 and EEM2), the uncertainty of the mean estimate from the three reference compounds does not include the variability between the single
reference compound estimates. When this variability is included (mean global EEM1 and EEM2; Table 2) the estimated uncertainty, as expected, increases. Furthermore, the error terms for the estimates derived from single reference compounds are based only on uncertainties of the literature derived emissions estimates for these compounds (may or may not include fully propagated errors) and the error terms of the measured ERs. It is not possible to determine the additional error associated
with deriving these global Hg emissions estimates from ERs measured in only one biome, which would likely lead to much higher uncertainties.

The limited availability of atmospheric Hg (either GEM/TGM or combined GEM, GOM, and PBM) measurements made in biomass burning plumes have also resulted in high uncertainties in emissions estimates made by more complex modelling efforts. Friedli et al. (2009) used biome specific EFs to
estimate global Hg biomass burning emissions. Yet, the EFs specific to each biome were based off highly uncertain soil-based estimates (change in soil Hg concentration before and after fire), simply "guesses", or by converting ERs (many from sites distant from source) to EFs based on the ratio of these two variables ([GEM:CO ER] / [GEM EF]) in the Washington State fires that we have shown incorporates elevated uncertainty related to their assumed ratio of carbon contaminant emissions





(Friedli et al., 2009). They estimated $675 \pm 240$ Mg yr$^{-1}$ (or between $708 - 1350$ Mg yr$^{-1}$ based a single, non-biome specific EF scenario) of Hg emissions from global biomass burning (Friedli et al., 2009). When considering the error term of this estimate, it would likely be much higher were it to include the fully propagated error of all these highly uncertain EF values and the assumptions made in their derivation. A recent effort produced a global TAM (an assumed 15 % PBM fraction was

added to the GEM concentrations) emissions estimate of 400 Mg yr$^{-1}$ (uncertainty described as "large") using a transport and transformation model (De Simone et al., 2017). They also assumed a single TAM:CO ER based on the mean of all studies that have measured Hg in plumes (De Simone et al., 2017). Their work did highlight the importance of including data inputs from different biomes in a global estimate, be that from either a combined mean value from the different biomes or a value

for each biome. At any rate, many of these TAM:CO ERs included in their assessment were measured at receptor sites distant from fire sources, which, as we have discussed, may overestimate this value due to potential difference in the atmospheric residence times of TAM and CO.

An additional uncertainty is the assumed fraction of PBM that we made no measurements of in the GLP fire. All our Hg emissions estimate methods indicate Hg emissions increase proportionally to

the assumed PBM concentrations increases (Table 2). However, this is not the case in more complex models that integrate transport and atmospheric chemistry processes. PBM has a much shorter lifetime than GEM and deposits much nearer to sources; increasing the PBM fraction leads to greater inputs of Hg into local and regional terrestrial matrices (De Simone et al., 2017; Fraser et al., 2018). Thus, it is imperative we better constrain our knowledge of Hg speciation in biomass burning

emissions via in plume measurements of GEM, GOM, and PBM. This has particular importance from a global Hg biogeochemical cycling standpoint as both De Simone et al. (2017) and Fraser et al. (2018) have shown substantially increased Hg deposition during simulations with elevated PBM inputs (compared to those without PBM emissions) in their global and Canadian transport and fate models, respectively.

## 3.4 GLP Fires Emissions Estimate Using TERRA

The GEM concentration screen for screen 1 of the flight generated from the TERRA algorithm and simple Kriging interpolation is displayed in Fig. 5. Only the emissions transfer rate of the south plume was considered in the TERRA-based emissions estimates as the concentration data are additive in this algorithm. Including the north plume would overestimate emissions regardless of whether the

north plume was derived from a separate undetected fire (i.e. not part of the GLP fire burned area) or resulting from the changing overnight winds (counting emissions from the GLP fires twice). The measured 2-min ($0.766 \pm 0.153$ kg hr$^{-1}$) and estimated 2-sec ($0.68 \pm 14$ kg hr$^{-1}$) GEM concentration data gave similar results and the TERRA emissions estimates discussed here are based on the



measured 2-min value to allow directly comparable data to the other emissions estimates. Assuming
a constant GEM TERRA-derived emission transfer rate across Screen 1 over the whole burning
period of the GLP fires (72 hrs) gives an emissions estimate of $104 \pm 20.9$ kg of GEM for the GLP
fires. Nonetheless, the MODIS satellite imagery shows the fires peaked on the day of the flight ($25^{th}$
of June); and hence, this assumption creates a large overestimation of the emissions estimate based
on the whole fire. To account for changes in the fire intensity, the emissions transfer rate was adjusted
by the number of MODIS fire and thermal anomalies observed each day and night (see Section S7
for fire and thermal anomaly data) resulting in an improved estimate of $22.0 \pm 7.3$ kg of GEM for the
GLP fires, which is remarkably similar to EEM3 ($21 \pm 10$ kg). This error term includes the 26.6 %
uncertainty associated with the MODIS satellite fire characterisation (Freeborn et al., 2014). The
similarity between the TERRA estimate and the more widely used and largely empirically-derived
EEM3 estimate for the GLP fires gives weight to the versatility of this algorithm that has only been
previously used to assess industrial pollutant emissions (Gordon et al., 2015; Liggio et al., 2016).
Future studies monitoring pollutant emissions from biomass burning using aircraft would benefit
from the inclusion of TERRA in their assessment.

## 4   Conclusions and Recommendations

This study presents a robust dataset describing elevated GEM concentrations in a near-source biomass
burning emissions plume using empirical relationships between GEM and reference contaminants
($CO$, $CO_2$, and $CH_4$). These data are the most constrained (lowest uncertainty) of any experimental
study measuring GEM concentrations and emissions in biomass burning plumes. The measured GEM
enhancements, ERs (for multiple reference compounds), and EFs provide a valuable contribution to
the literature on Hg emissions from biomass burning. We were able to derive a robust GEM emissions
estimate of $21 \pm 10$ kg from the GLP fire using the empirically calculated EFs that is well supported
by the $22 \pm 7$ kg emissions estimate using the TERRA algorithm. Neither of these estimates require
external data inputs (literature values) of reference compounds nor extensive assumptions.

Nonetheless, upscaling these emissions to all boreal and global forest fires is inherently problematic,
a point we have stressed in detail. The broad range of emissions estimates made for boreal and global
forest fires highlights uncertainty associated with factors such interannual variability in burned area
and differing vegetation types. Another major source of error is the calculation of emissions estimates
using data from a single reference compound, a concern that has been somewhat neglected by the
atmospheric Hg community. Typically, ERs or EFs have been based on solely $CO$ (or occasionally
$CO_2$) and used to estimate Hg emissions from biomass burning. These calculations are generally
based on very limited empirical data often without a complete description of their uncertainty. We
stress potential error associated with variable $CO$ enhancements between different fires (vegetation





type and fire intensity) and contrasting atmospheric lifetimes of these two contaminants applied in these methods. Similarly, Hg ERs with other potential reference compounds (i.e. $CO_2$, CH4, and NMHC) have their own inherent uncertainties.

This does not mean that the Hg ERs should not be used, only that their caveats be fully described, and methods developed to reduce these uncertainties. Help may be on its way; a recent publication attempts to use a statistical modelling approach that combines multiple tracers or reference compounds to predict emissions (Chatfield and Andreae, 2020). Future efforts modelling Hg emissions from biomass burning are likely to benefit from broader approaches such as this. Additionally, more near source monitoring of Hg emissions from biomass burning, particularly using aircraft-based measurements of the different Hg species (GEM, GOM, and PBM) and carbon co-contaminants (CO, $CO_2$, and $CH_4$), across all biomes would assist in narrowing the uncertainty of Hg based ERs and potentially produce ERs applicable to vegetation type.

## Author Contribution

D.M. was on the flight managing gas measurement instruments including the Picarro instruments and Tekran 2537X, managed the Tekran 2537X operation and maintenance throughout the Oil sands Monitoring Program (OSMP), created Figs. 1-4 and all tables within the paper, managed the calculations of ERs, EFs, and emissions estimates, and wrote the paper; G.S. was in charge of the technical setup of the Tekran 2537X and ensuring it was fully operational and quality controlled for aircraft use in the OSMP, assisted with technical difficulties during the OSMP, and contributed to revisions of the manuscript; A.D. was responsible for the TERRA modelling, created Fig. 5 contributed to Fig. 1, assisted with running Tekran 2537X instrument during the OSMP, and contributed to revisions of the manuscript; K.H. was project co-leader of the OSMP, contributed the CO, $CO_2$, and $CH_4$ data (collection, QA/QC, and analyses), and contributed to revision of the manuscript and ER, EF, and emissions estimate calculations; A.S. was responsible for overall project planning and management for the Hg component of this project and contributed to revisions of the manuscript.

## Acknowledgements

The authors would like to acknowledge the entire team of our skilled technicians, ground maintenance staff, pilots, administration, and scientists from the AQRD of ECCC and the NRC working on the OSMP. Special thanks to Richard Mittermeier and John Liggio for their contribution of CO, $CO_2$, $CH_4$, and NMHC data collection and QA/QC and to Shao-Meng Li and Stewart Cober for their tireless work in bringing the aircraft component of the OSMP project into fruition. This work was primarily funded by Environment and Climate Change Canada. Additionally, the authors acknowledge the vital



data on the wildfire provided by Sindy Nicholson from the Wildfire Management Branch of the Government of Saskatchewan that, in particular assisted greatly with the determination of the burned area of the GLP fires. D.M. would like to thank Prof. Dr. Meinrat O. Andreae from Max Planck Institute of Chemistry in Mainz, Germany for his invaluable help ensuring the emissions estimate

calculations were correct. D.M. also acknowledges support provided through the National Sciences and Engineering Research Council of Canada (NSERC) Postdoctoral Fellowship Program and his supervisor Prof. Dr. Harald Biester at the Technical University of Braunschweig for allowing time to finalise this project.

## Competing Interests

The authors declare that they have no conflict of interest.

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





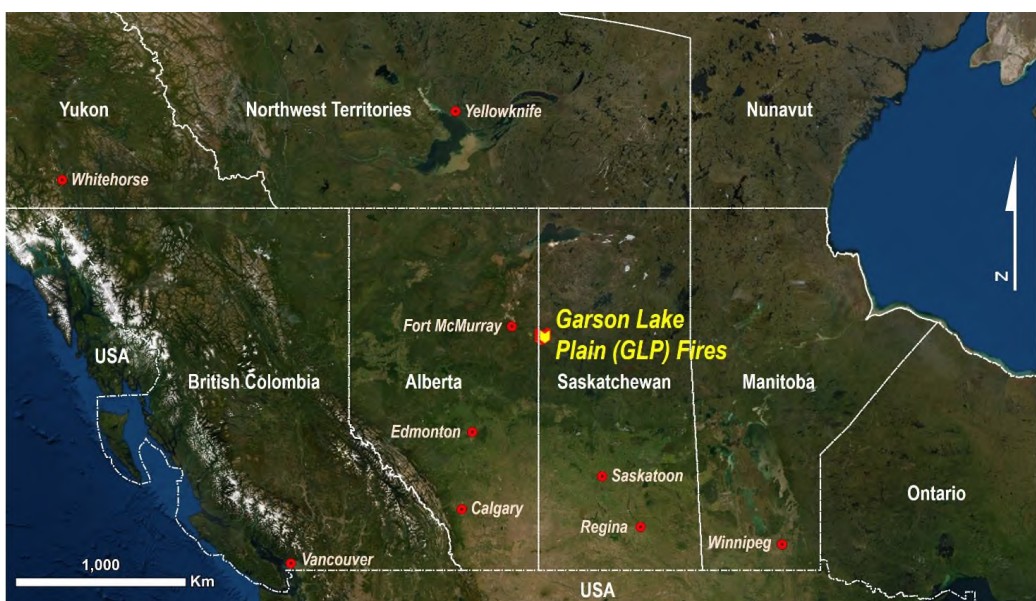

*Figure 1: Regional map showing Garson Lake Plain (GLP) fires location in Northern Saskatchewan, Canada (© ArcGIS; ESRI).*

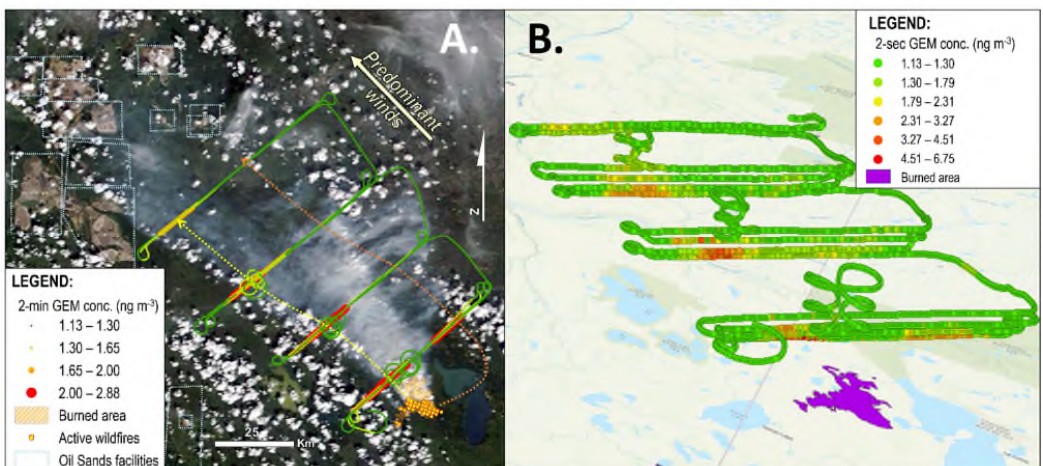

*Figure 2: Panel (a) shows the 2-minute measured GEM concentrations along the flight track, overlaid*

*onto the satellite image of the wildfire taken from MODIS Satellites at approximately 18:59 GMT on the 25th of June 2018 (near end of flight) (NASA, 2020). Yellow and orange dotted lines in Panel (a) show approximate path of the south and north plumes, respectively. Panel (b) shows the 2-second GEM concentration calculated by conversion of the 0.5 Hz CO data using the GEM:CO emissions ratio (ER) along the flight path in three-dimensions.*


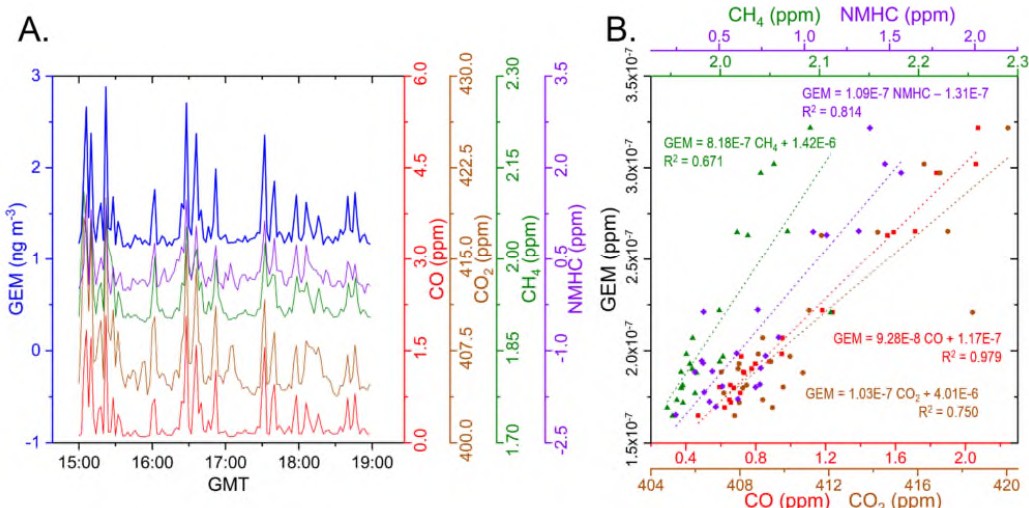

*Figure 3: Panel (a): Concentrations of GEM (2-min measured), and mixing ratios of CO, CO₂, CH₄ and NMHCs during fire monitoring flight. Panel (b): Mixing ratio regressions of GEM against CO, CO₂, and CH₄ during the wildfire monitoring flight (this data is based on only the GEM data elevated*
*>125 % of the background concentration); ERs are derived from the slopes of these regressions.*

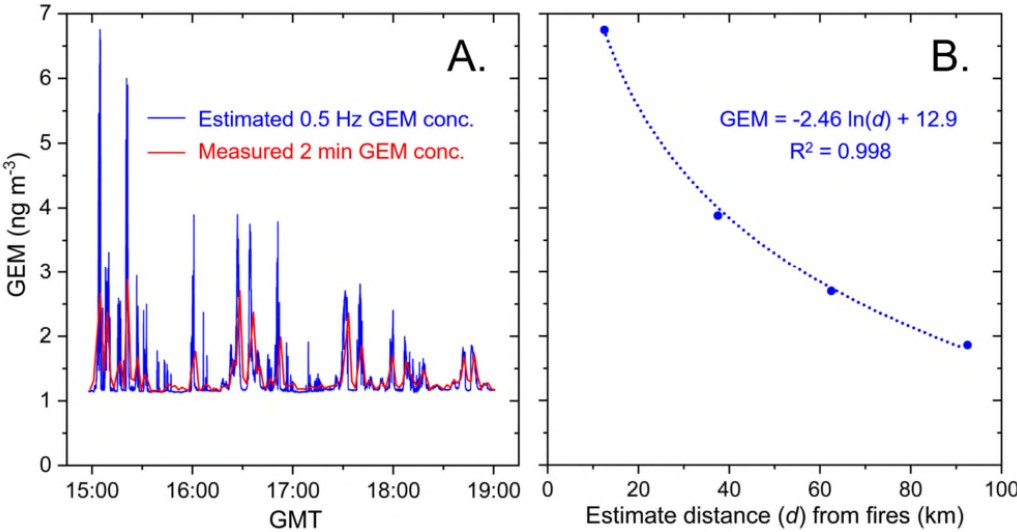

*Figure 4: Panel (a): 2-min measured and 2-sec calculated GEM concentration; the latter was calculated by conversion of the 0.5 Hz CO data using the GEM:CO emissions ratio (ER) measured*
*in the GLP fires. Panel (b): The maximum 2-sec calculated GEM concentration derived from GEM:CO ER for each screen and the estimated distance this measurement was from the GLP fires.*

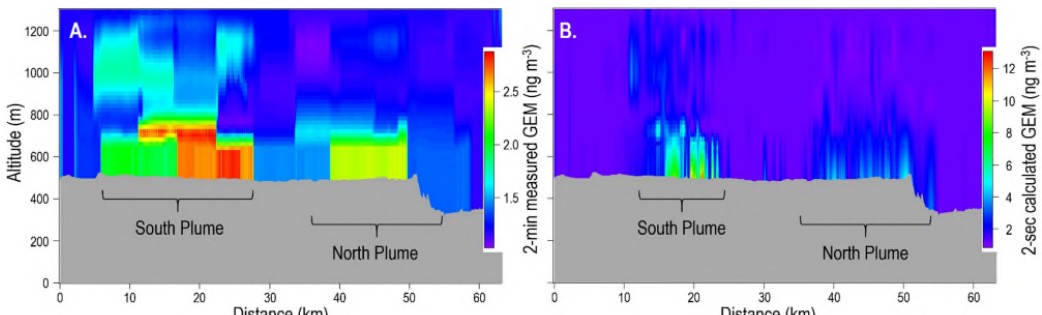

*Figure 5: Simple Kriging interpolation of TERRA GEM concentration screen for Screen 1 of the GLP*
*fires. Panel (a) is based on the 2-min measured GEM concentration data. Panel (b) is based on the*
*2-second GEM concentration calculated by conversion of the 0.5 Hz CO data using the GEM:CO*
*emissions ratio (ER). Note concentration differences between the 2-min and 2-sec GEM*
*concentration data in the figure legends.*

*Table 1: Enhancements, ERs, and EFs of GLP fire and the most comparable fires with near source*
*measurements of GEM.*

|  | **This study** | **Brunke et al. (2001)** | **Friedli et al. (2003a)** | **Friedli et al. (2003b)** |
|---|---|---|---|---|
| **Location** | NW Saskatchewan, Canada | Cape Point, South Africa | N Ontario, Canada | Washington State, USA |
| **Vegetation Type** | Boreal forest | Fynbos shrubland | Boreal forest | Temperate forest |
| **Max. measured GEM enhancement** | ≈2.4x | ≈0.45x | ≈0.4x | ≈6x |
| **GEM:CO** | $9.28 \pm 0.29 \times 10^{-8}$ | $2.1 \pm 0.2 \times 10^{-7}$ | $2.04 \times 10^{-7}$ * | $6.7 \pm 0.4 \times 10^{-8}$ |
| **GEM:CO$_2$** | $1.03 \pm 0.13 \times 10^{-8}$ | $1.2 \pm 0.3 \times 10^{-8}$ | $1.49 \pm 0.22 \times 10^{-8}$ | - |
| **CO:CO$_2$** | $0.111 \pm 0.013$ | $0.055 \pm 0.001$ | $0.10 \pm 0.02$ | - |
| **GEM:CH$_4$** | $8.2 \pm 1.2 \times 10^{-7}$ | - | - | - |
| **GEM:NMHC** | $1.09 \pm 0.11 \times 10^{-7}$ | - | - | - |
| **EFs (µg kg$^{-1}$)** | $99 \pm 25$ | - | $112 \pm 30$ ^ | $108 \pm 57$ |

* Value taken from the supplementary information of Friedli et al. (2009) – no uncertainty given.

^ Uncertainty of this estimate was recalculated to include their measured 20 % variability in the ratio of CO:CO$_2$.

All values include one extra significant digit to reduce rounding errors for any subsequent calculations (where possible).



*Table 2: Emissions estimates of Hg from biomass burning based on the three emissions estimate methods, three reference contaminants, and four PBM fraction scenarios described in the methods section.*

| | EEM1 - Literature emissions adjusted for measured ERs | | | | EEM2 - Literature EFs adjusted for measured ERs | | | | | | | EEM3 - Measured EFs and ERs | |
|---|---|---|---|---|---|---|---|---|---|---|---|---|---|---|
| **Hg emissions from global fires (Mg yr⁻¹)** | | | | | | | | | | | | | | |
| Reference pollutant | CO (28%) | | $CO_2$ (22%) | | $CH_4$ (30%) | | **Mean Global EEM1 (49%)** | | CO (58%) | | $CO_2$ (46%) | | $CH_4$ (64%) | | **Mean Global EEM2 (55%)** | | **Global EEM3 (51%)** | |
| Hg Scenario | value | error | value | error | value | error | value | error | value | error | value | error | value | error | value | error | value | error |
| 0% PBM | 211 | 61 | 355 | 79 | 132 | 39 | 233 | 113 | 660 | 380 | 590 | 270 | 460 | 300 | 570 | 320 | 810 | 410 |
| 3.8% PBM | 220 | 63 | 369 | 82 | 137 | 41 | 242 | 117 | 690 | 400 | 610 | 280 | 480 | 310 | 590 | 330 | 850 | 430 |
| 15% PBM | 249 | 72 | 417 | 93 | 155 | 46 | 274 | 133 | 780 | 450 | 690 | 320 | 550 | 350 | 670 | 370 | 960 | 480 |
| 30% PBM | 302 | 87 | 507 | 112 | 189 | 56 | 332 | 161 | 940 | 550 | 840 | 380 | 660 | 420 | 810 | 450 | 1160 | 590 |
| **Hg emissions from boreal forest fires (Mg yr⁻¹)** | | | | | | | | | | | | | | |
| Reference pollutant | CO (79%) | | $CO_2$ (78%) | | $CH_4$ (80%) | | **Mean Boreal EEM1 (79%)** | | CO (86%) | | $CO_2$ (78%) | | $CH_4$ (90%) | | **Mean Boreal EEM2 (85%)** | | **Boreal EEM3 (81%)** | |
| Hg Scenario | value | error | value | error | value | error | value | error | value | error | value | error | value | error | value | error | value | error |
| 0% PBM | 19.3 | 15.3 | 29 | 23 | 11.4 | 9.1 | 19.9 | 15.7 | 14.7 | 12.7 | 13.1 | 10.3 | 9.5 | 8.5 | 12.7 | 10.8 | 18.2 | 14.8 |
| 3.8% PBM | 20.0 | 15.9 | 30 | 24 | 11.8 | 9.4 | 20.7 | 16.3 | 15.3 | 13.2 | 13.7 | 10.7 | 9.9 | 8.8 | 13.2 | 11.2 | 18.9 | 15.4 |
| 15% PBM | 22.7 | 18.0 | 34 | 28 | 13.4 | 10.7 | 23.4 | 18.5 | 17.3 | 14.9 | 15.5 | 12.1 | 11.2 | 10 | 15.0 | 12.7 | 21.4 | 17.4 |
| 30% PBM | 28 | 22 | 42 | 33 | 16.2 | 12.9 | 28 | 22 | 21.1 | 18.1 | 18.8 | 14.7 | 13.6 | 12.1 | 18.1 | 15.4 | 26 | 21 |
| **Hg emissions from GLP fires (kg)** | | | | | | | | | | | | | | |
| Reference pollutant | CO (-%) | | $CO_2$ (-%) | | $CH_4$ (-%) | | **Mean GLP EEM1 (-%)** | | CO (58%) | | $CO_2$ (46%) | | $CH_4$ (65%) | | **Mean GLP EEM2 (56%)** | | **GLP EEM3 (51%)** | |
| Hg Scenario | value | error | value | error | value | error | value | error | value | error | value | error | value | error | value | error | value | error |
| 0% PBM | - | - | - | - | - | - | - | - | 16.6 | 9.7 | 14.8 | 6.8 | 11.6 | 7.6 | 14.4 | 8.0 | 20.6 | 10.4 |
| 3.8% PBM | - | - | - | - | - | - | - | - | 17.3 | 10.1 | 15.3 | 7.1 | 12.1 | 7.9 | 14.9 | 8.4 | 21.4 | 10.9 |
| 15% PBM | - | - | - | - | - | - | - | - | 19.6 | 11.4 | 17.4 | 8.1 | 13.7 | 8.9 | 16.9 | 9.5 | 24.2 | 12.3 |
| 30% PBM | - | - | - | - | - | - | - | - | 23.8 | 13.9 | 21.2 | 9.8 | 16.6 | 10.7 | 20.5 | 11.5 | 29.4 | 15.0 |

· Values in parenthesis next to reference contaminants are the coefficient of variation % (CV%) for that set of estimates.
· Emissions from Garson Lake Plain (GLP) fires are in different units (kg).
· All estimates and error terms include one extra significant digit to reduce rounding errors for any subsequent analysis.
· EFs – emissions factors; ERs – emissions ratios.