# Peer review of "Where there is smoke there is mercury: Assessing boreal forest fire mercury emissions using aircraft and highlighting uncertainties associated with upscaling emissions estimates."

_Atmospheric Chemistry and Physics, 2020_

## Referee Comment (RC1) · Anonymous Referee #1 · 11 Jan 2021

**General Comments**

The manuscript estimates Hg emissions from wildfires using different methods based on aircraft measurements of GEM, CO, CO2, CH4 and NMHCS and also assesses the uncertainty of these methods. The manuscript provided thorough data analysis and the work would make important contribution to study Hg emission from biomass burning. However, my major complain is the structure of the manuscript and the presentation of tables and figures, which makes it hard to read and go through. Some paragraphs are really long, and it is hard to process the information. It would be helpful to break them

down when possible. Please also see the following section for detailed comments.

Specific comments:

1. Line 127: what are NNW and NNE? please spell them out.

2. Figure 1: what are the red circles for? Please explain them.

3. Pleas change the label A and B to (a) and (b) to be consistent with the figure captions from Figure 1 to Figure 5.

4. Line 300 - 301: it is not clear how the Hg emissions estimates are based on CO, CO2 and CH4 as reference compounds.

5. Figure 3 is a little hard to read with different axis for different species. Is it possible to normalize the numbers somehow and just use one axis so that the figure is more readable?

6. Section 2.4 includes so much information and it would be helpful to break it down or better organize it. For example, one section can discuss in general how Hg emissions from biomass burning are estimated from ERs and EFs, which provides the background information. Then, another section specifically describes the four different methods used in this work. It is also helpful to make a table to list the four different methods used in this work which would make it easier to see the difference and similarity. It is hard to go through so much text information.

7. Line 238: how is 125% decided?

8. Section S6: The table should be moved to the main text rather than in the supplement as it is part of the method section.

9. Table 2 is hard to read. It would be helpful to break it down to three tables.

Corrections:

Line 45: inspiration? You mean respiration?

Line 220: CO2 and CH4, make sure the subscript formats are correct

Line 257: the * symbol is missing in equation (3)

---

## Referee Comment (RC2) · Anonymous Referee #2 · 29 Jan 2021

The authors use air craft observations and statistical methods to derive new estimates for mercury emission factors from wildfires. The presented work is of importance as natural mercury (re-)emissions become ever more relevant with the ongoing reduction of anthropogenic emission as internationally agreed under the Minamata Convention on mercury. Forests and vegetation are a major storage for mercury. With increasing temperatures wild fires will become ever more prevalent (Veira et al., 2016 ). The year 2020 with its massive fires in the Arctic were a recent reminder of this.

The presented manuscript is generally well written. However it is a bit long and the abstract needs some language editing. The employed methodologies are sound and the authors show an extensive knowledge of the field. They compare their results with previous estimates and what I find the most important clearly state and quantify the uncertainties of their method.

I support publication of the manuscript after a few questions have been answered.
Finally, I need to apologize for taking so long with this review.

**Remarks**

p.6
Section 2.2: How sure are you that you measure GEM only and not GEM + a fraction x of GOM? Given that you heat the sample line from ambient temperature to 50°C I think x could be larger than zero. What is your oppinion on this?

p.7
Section 2.3/2.4: I wonder what the detection limit (dl), especially for the CO measurements is. As there are large differences in the detection limits of background CO instruments and regular ones. The Picarro G2401-m manual I could find online states a dl for CO of 0.2 ppm which seems low compared to the observed CO average of 0.134ppm. Please correct me if I am wrong.
Anyhow, I would appreciate if you added this information.

p.9
ll.283-286: Do you include small scale fires from GFEDv4? Do you think small scale or smoldering fires could be of relevance to Hg emission from wildfires?

p.13
ll.392-402: You compare your results to GEM:CO emission ratios from the literature. To what extend do you expect that the mercury content of the fuel burned during each of these campaigns is comaparable? Or shorter: Could differences in Hg content explain the variability of the different ERs?
*I see you discuss this later in the manuscript but maybe it makes sense to mention it here briefly.*

ll.396: 'This places the GEM:CO ER determined in our study near the lower end of this range, ...'
Please add a reference to Table 1 here. Otherwise one has to search for it, readers will surely thank you.

p.14
ll.430-434: What effects the $CO:CO_2$ ER the most? I would expect temperature und thus fire size besides the fuel type could be important?

p.20
ll.628-640: I suggest that you give your best guess

**Language**

p.1
l.12: please correct: 'the contaminant to the atmosphere'

l.13: I suggest 'part' in stead of 'compartment'

l.17: 'emissions plume' sounds odd, I suggest just 'plume'

l.19: '…,which ist 2.4 times background concentration'

p.14
ll.424-430: Text formatting seems off here.

**References**

Veira, A., Lasslop, G., Kloster, S. 2016. Wildfires in a warmer climate: Emission fluxes, emission heights, and black carbon concnetrations in 2090-2099.

---

## Referee Comment (RC3) · Anonymous Referee #3 · 1 Feb 2021

The authors present aircraft measurements of GEM, CO, CO2, CH4, and NMHCs in the plume of a forest fire in Saskatchewan. From these data they derive the emission ratios and calculate GEM emissions by three different methods. In addition, they calculate GEM flux using the screen flown downwind of the fire. The results of GEM emissions calculated by different methods are compared and their uncertainties discussed.

The paper is well written but for publication it needs to provide more information at times, mentioned below. Some questions, listed below should also be answered:

The purpose of the extensive discussion of the GEM enhancements is not clear. In addition, it will strongly depend on the meteorology which is omitted from the discussion. Without consideration of the windspeed (dilution) and the distance to the fire, the comparison with measurements published by other authors does not make much sense.

Section 2.1: Measurement of wind speed and direction onboard aircraft is not easy. The reader would like to know how these parameters were measured and with which uncertainties. This information is needed to assess the uncertainty of fluxes calculated by TERRA.

Sections 2.2 and 2.3: What are the estimated uncertainties of the individual GEM, CO, CO2, and CH4 measurements. These uncertainties are needed for assessment of the quality of ERs: e.g. the poorer quality of GEM/CH4 ratio could be caused by higher uncertainty of CH4 measurements? They are also needed for the orthogonal correlations (Cantrell, ACP, 8, 5477-5487, 2008).

Line 233: I presume the GEM background is given as an average of 2 min measurements. What was the number of the GEM measurements used in this average?

Lines 237-241: The consideration of only GEM enhancements >125% is probably not justified for several reasons: a) It is arbitrary – why not 115%? b) The selection of >125% GEM data may show only a part of the plume which may not be representative of the whole plume. c) With increasing distance to the fire, the section of plume with >125% would decrease relatively to the whole plume which again poses the question of representativity. d) The authors state that the data below 125% the enhancement are "too variable and too uncertain" to be considered. The concern about the uncertainty should not be the problem if the authors used orthogonal regression with uncertainties of both GEM and X. The variability should also be no problem: the more points the smaller R is significant. Reference to Yokelson et al. (2013) is not quite appropriate for the situation here, i.e. with measurements up to 100 km distance from the fire with

nearly constant background mixing ratios.

In addition, the limitation to GEM enhancements >125% even seems to be unnecessary: the ERs in figure S4.1 calculated from all data and in Table 1 calculated with only GEM enhancements >125% are probably the same in statistical terms, i.e. cannot be distinguished taking into account the ERs uncertainties and the number of measurements. An additional table of ERs from Table 1 and ERs from figure S4.1 could be used to illustrate the necessity of the >125% threshold or its absence.

Paragraph lines 245-248: What type of regression was used: LSQF or orthogonal one? The usual LSQF should not be used to calculate ERs!!! The last sentence is difficult to understand because any type of regression automatically adjusts for the backgrounds – it is just a shift in the coordinate system. Probably CO, CO2, CH4, and NMHC measurements were converted to 2 min averages synchronized with 2 min GEM for correlations? If so, it should be mentioned.

Paragraph lines 249-252: How was the integration made? Only for the enhancements >125%?

Paragraph lines 305-327: More information is needed in the description of TERRA calculation? How were GEM measurements (2 min) interpolated? The treatment of the layer above the highest transection and the inversion layer has to be mentioned too. Some of this information is provided in Section 3.4 but the reader would expect it here.

Lines 348 and 349: How was the correlation made: orthogonal? 2min data? What are the regression lines?

Paragraph lines 459-464: The conversion of 2 min GEM measurements into 0.5 Hz data using the GEM/CO ratio was probably made mainly for the TERRA calculations. If so, it should be mentioned.

Paragraph 465-480: The text here is highly speculative because the estimations are

[Figure]

strongly dependent on the meteorology which is not mentioned. In addition, it does not contribute much to the purpose of the paper.

Table 1: The comparison of maximal measured GEM enhancements is strongly dependent on their temporal resolution, as shown in this work, and on the meteorological parameters (especially wind speed in combination with the distance to the fire, i.e. dilution). Without taking all these factors into account, the comparison does not make much sense and as such should be deleted from the table, and also from the text.

Table 2: "Uncertainty" should be used instead of "error" here and throughout the text. The calculation of uncertainties and the terms used in the equation 7 should be given either in the manuscript or in the supporting information.

Fig. 2b: I wonder about 2s GEM data derived from 0.5 Hz CO data using the GEM/CO ratio. What is this conversion good for? I find its presentation misleading because it pretends much higher density of GEM data than available. The 0.5 Hz CO, $CO_2$, and $CH_4$ measurements should be converted to the 2 min GEM time stamp, at least for the regressions.

Fig. 3: Which type of regression was used? The usual one (LSQF) or one which considers the X and the Y uncertainties? What was the number of correlated points: R without the number of points does not say anything about the significance of the regression. Dtto the figure S4.1.

Fig. 5: "Distance" in the name of x axis may be easily mixed up with the distance to fire.

---

## Author Comment (AC1) · 23 Feb 2021

R1-C1. General Comments The manuscript estimates Hg emissions from wildfires using different methods based on aircraft measurements of GEM, CO, CO2, CH4 and NMHCS and also assesses the uncertainty of these methods. The manuscript provided thorough data analysis and the work would make important contribution to study Hg emission from biomass burning.

We thank the reviewer for their positive assessment of the manuscript. Please note

that after discussion between the co-authors NMHCs was updated to NMOGs (non-methane organic gases) as the measurements also include oxidised organic gases; and hence this is the more correct terminology.

R1-C2. However, my major complain[t] is the structure of the manuscript and the presentation of tables and figures, which makes it hard to read and go through. Some paragraphs are really long, and it is hard to process the information. It would be helpful to break them down when possible. Please also see the following section for detailed comments.

We appreciate the comment and have attempted to break up some of the longer paragraphs (on approximately nine occasions in total).

Specific comments: R1-C3. Line 127: what are NNW and NNE? please spell them out.

These are very well used direction abbreviations. North of North East (NNE) and North of North West (NNW). We believe both are well established and it is quite clear in the text that they are referring to direction. We don't believe an update is necessary here.

R1-C4. Figure 1: what are the red circles for? Please explain them.

These refer to the cities of Canada. They are adjacent to their city names as is common in maps. We have added the following to the end of Fig1 caption: "..., Canadian Provinces (white dashed lines), and major/relevant cities (red dots)...".

R1-C5. Pleas[e] change the label A and B to (a) and (b) to be consistent with the figure captions from Figure 1 to Figure 5.

The labels for all composite figures were corrected as recommended.

R1-C6. Line 300 - 301: it is not clear how the Hg emissions estimates are based on CO, CO2 and CH4 as reference compounds.

Thank you for the comment and we agree that this could be made clearer. Two updates to the text were made in the revised manuscript to clarify this: Lines 310-313: "Emissions estimate method 2 (EEM2) coverts a literature derived EF for a reference compound (see Section S7 and Andreae, 2019 for the EF values used) to a Hg EF using the molecular weight of each species and the measured ER between GEM and the reference compound based on Equation 3. The emission estimate (Qx) is then calculated according to Equation 5:"

Lines 318-319: "EEM2 makes a separate Hg emissions estimates based for each reference compound used (CO, CO2, and CH4)."

R1-C7. Figure 3 is a little hard to read with different axis for different species. Is it possible to normalize the numbers somehow and just use one axis so that the figure is more readable?

While we understand there is a lot of information in this graph, we deem it to be very important to the study (arguably the most important) and do not deem it appropriate to normalise the axes. Normalising the data (as suggested) would make the data difficult to interpret (not in their original form) and cause the data to overlap too much. The axes ranges have been specifically controlled so the curves and plots of each relationship impact the other data (overlap) as little as possible. Much time and consideration were spent in designing these figures for optimal clarity and understanding of the reader. We believe presenting the data like this is the most appropriate way to present different species in the one graph for comparison. Separating either Panel (a) or Panel (b) into individual graphs for each species would result it much more difficult comparisons for the readers and excessive use of space. Furthermore, using multiple axes to represent different species is a common approach used in atmospheric chemistry.

R1-C8. Section 2.4 includes so much information and it would be helpful to break it down or better organize it. For example, one section can discuss in general how Hg emissions from biomass burning are estimated from ERs and EFs, which provides the background information. Then, another section specifically describes the four different

methods used in this work. It is also helpful to make a table to list the four different methods used in this work which would make it easier to see the difference and similarity. It is hard to go through so much text information.

This is an excellent recommendation, and the section has now been divided into three. 2.4 Emissions Ratio; 2.5 Emissions Factors; 2.6 Emissions estimates. Former lines 265 - 277 were also moved up to the end of new section 2.4 (Lines 266-278 in the revised manuscript).

R1-C9. Line 238: how is 125% decided?

The text has been modified to clarify how we came to this value. This value was selected based on a sensitivity analysis that compared issues of variable background concentrations of reference compounds (low cut-off value: >1.1x background) against having limited data and higher uncertainty (higher cut-off value: >1.5x background). This sensitivity analysis is now included in the supplementary information (Section S5). This has also been clarified with an updated statement (Lines 243-252) in the revised manuscript as follows:

"All emissions ratios (ERs) (and subsequent emissions factors and emissions estimates calculations) are based on GEM concentrations that were enhanced by >1.25x background GEM concentration (>1.47 ng m-3). Data below this fraction were more variable and uncertain and included concentration values below background for some of the reference compounds, particularly for the $CO_2$ enhancements due to the more elevated and variable background concentration of $CO_2$ (Yokelson et al., 2013; Andreae, 2019). In total, 24 GEM concentration measurements were enhanced by >1.25x background. Increasing this cut-off value leads to a reduction in data and increased uncertainty in ERs (and emissions factors and emissions estimates). We believe the data cut-off >1.25x GEM background provides appropriate balance between the uncertainties of variable background values and reduced data. A sensitivity analysis of this value is assessed in Section S5."

R1-C10. Section S6 [Now Section S7]: The table should be moved to the main text rather than in the supplement as it is part of the method section.

Our preference is to keep this table in the supplementary section. While the emissions factors are indeed important, they are a calculation step in reaching an emissions estimate and complementary information. In contrast, emissions ratios can be used for other functions (i.e. ERs can be used as identifiers of forest fire plumes) and are thus in the main text. Emission factors are discussed in the text, where required, and the "best estimate" based on EEM3 (using minimal possible literature inputs) is listed in Table 1 of the manuscript (99 $\pm$ 26 $\mu$g kg-1). If the reader requires further details, they can easily refer to the SI, which is appropriately referenced in the manuscript. Additionally, we acknowledge this is a long manuscript and are reluctant add more length.

R1-C11. Table 2 is hard to read. It would be helpful to break it down to three tables.

We thank the reviewer for their recommendation and Table 2 has been divided into three: (a) global fire emissions; (b) boreal forest fire emissions; and (c) GLP fire emissions, in the revised manuscript.

Corrections: R1-C12. Line 45: inspiration? You mean respiration?

Inspiration is what we mean. The act of inspiring air or the drawing in of air during respiration.

R1-C13. Line 220: $CO_2$ and $CH_4$, make sure the subscript formats are correct

Corrected.

R1-C14. Line 257: the * symbol is missing in equation (3)

Corrected.

Anonymous Referee #2

R2-C1. The authors use aircraft observations and statistical methods to derive new

estimates for mercury emission factors from wildfires. The presented work is of importance as natural mercury (re-)emissions become ever more relevant with the ongoing reduction of anthropogenic emission as internationally agreed under the Minamata Convention on mercury. Forests and vegetation are a major storage for mercury. With increasing temperatures wild fires will become ever more prevalent (Veira et al., 2016). The year 2020 with its massive fires in the Arctic were a recent reminder of this. The presented manuscript is generally well written. However, it is a bit long and the abstract needs some language editing. The employed methodologies are sound, and the authors show an extensive knowledge of the field. They compare their results with previous estimates and what I find the most important clearly state and quantify the uncertainties of their method. I support publication of the manuscript after a few questions have been answered.

We greatly appreciate the positive sentiments and support for the work. We too feel it is a very important and contemporary topic. We do understand the manuscript is quite long, but it is our intention to be very holistic in our assessments and ensure all uncertainties are fully characterized to potentially improve the direction of study in this field in the future. To do this takes considerable space and we appreciate the consideration of this fact by both reviewer 1 and reviewer 2.

R2-C2. Finally, I need to apologize for taking so long with this review.

We totally understand given the current situation.

Remarks R2-C3. p.6 Section 2.2: How sure are you that you measure GEM only and not GEM + a fraction x of GOM? Given that you heat the sample line from ambient temperature to 50°C I think x could be larger than zero. What is your opinion on this?

We understand and anticipated reviewer questions in regard to this very important issue regarding the measured analyte. As such, we have already included a detailed discussion on exactly this in the supplementary information (Section S3). We will briefly summarize the main points as to why we firmly and empirically consider the target analyte GEM: 1. Yes, part of the sampling line was heated (4.5/5.44m), but part of it was not (0.95/5.44m). The non-heated line will likely remove a portion of GOM that will stick to the line (described in the methods section of the manuscript). 2. The setup employed a soda-lime trap with quartz wool plugs (intended primarily to remove moisture), both components are expected to remove a portion if not all GOM (see Lyman and Jaffe, 2012; Gustin et al., 2013; Slemr et al., 2016; 2018, and peer review comments of these papers). 3. The Teflon filters themselves remove a (small) portion of GOM as we have shown in an on-going study from work during atmospheric mercury depletion events at Alert in Arctic (conference proceedings: Stupple et al., 2019). 4. We have also confirmed that we are only measuring GEM during another flight in this field campaign while monitoring pollutant emissions from the Alberta Oil Sands. In this flight, we used a standard Teflon filter on the regular inlet line and a polysulphone cation exchange membrane (CEM) on the normal "zero-air" Channel. The CEM filters have been shown to allow GEM to pass at >99.9% efficiency, but remove GOM (Miller et al., 2019). By placing the CEM filter on the normal "zero-air" inlet we were able to alternately sample through these two different lines every 3rd (2-min) sample throughout the flight. There was no systematic difference between GEM concentration data measured by the two filter systems. Further, this flight showed evidence of minor GEM emissions from the facility, regardless of the sampling line that was used. Another flight at this same facility employed the standard setup (Teflon filter and no CEM) and also confirmed GEM emissions. Industrial facilities typically emit GOM (Carpi, 1997), and if the Teflon filter system sampled some GOM emitted from the facility then it would have showed systematically higher results than the CEM setup, but as I mentioned this was not the case. As a result of this evidence, we are confident that we measured GEM only during this flight. Please see Section S3 for further details.

R2-C4. p.7 Section 2.3/2.4: I wonder what the detection limit (dl), especially for the CO measurements is. As there are large differences in the detection limits of background CO instruments and regular ones. The Picarro G2401-m manual I could find online states a dl for CO of 0.2 ppm which seems low compared to the observed CO average

of 0.134ppm. Please correct me if I am wrong. Anyhow, I would appreciate if you added this information.

The Picarro G2401-m CRDS instrument has been successfully used to measure background and plume concentration levels of CO, CO2, CH4 and H2O down to low ppb sensitivity, in many measurement campaigns including on aircraft (e.g. Gordon et al., 2015; Baray et al., 2018; Liggio et al., 2019; Karion et al., 2013 – many of the same researchers involved in our monitoring campaign). From this study a 3x standard deviation value ($3\sigma$ value) of 12ppb was determined from 0.5 Hz measurements made in background air and this is the value we use as the instrument detection limit. This is similar precision to these aforementioned studies. Considering our background CO value of 0.134 ppm (which is more than 10x the $3\sigma$ value), we consider the instrument is certainly sensitive enough to measure these background CO mixing ratios. All data reported also went through three levels of quality assurance and control check. We present the CO data (and all other considered species) with high confidence. The following has been added to the revised manuscript (Lines 233-236) to describe the detection limits: "The $3\sigma$ values for CO, CO2, CH4, and NMOGs are 12, 380, 4, and 60 ppb, respectively, using the same approach described for the Tekran 2537X. These instrument uncertainties are similar to those described elsewhere (Gordon et al., 2015; Baray et al., 2018; Liggio et al., 2019; Karion et al., 2013) and these studies also outline the methods of these instruments in more detail."

R2-C5. p.9 ll.283-286: Do you include small scale fires from GFEDv4? Do you think small scale or smoldering fires could be of relevance to Hg emission from wildfires?

We only consider the fires included in the GFEDv4 data reported in Giglio et al. (2013). While this is a very interesting point raised by the reviewer and absolutely worthy of consideration in future modelling studies on Hg emissions from biomass burning, we think such a discussion goes beyond the scope of this manuscript (and would lengthen it further). Our focus is on measuring concentrations, establishing robust emissions ratios, estimating emissions and discussing issues that have been made previously

when upscaling such emissions. We already discuss the uncertainty associated with the interannual variability in burned area, and a consideration or small fires (or not) would generally fall under such an overarching uncertainty.

R2-C6. p.13 ll.392-402: You compare your results to GEM:CO emission ratios from the literature. To what extent do you expect that the mercury content of the fuel burned during each of these campaigns is comparable? Or shorter: Could differences in Hg content explain the variability of the different ERs? I see you discuss this later in the manuscript but maybe it makes sense to mention it here briefly.

The discussion on GEM:CO ERs is given in Lines 454-468 in the revised manuscript. The discussion on the influence of differing ERs caused by vegetation type is at the end of the next paragraph and into the paragraph following that. We believe the two are well connected in the text and follow the flow of the discussion and don't believe another comment is warranted as it will lead to repetition.

R2-C7. ll.396: 'This places the GEM:CO ER determined in our study near the lower end of this range, ...'Please add a reference to Table 1 here. Otherwise one has to search for it, readers will surely thank you.

This addition was made.

R2-C8. p.14 ll.430-434: What effects the CO:CO2 ER the most? I would expect temperature and thus fire size besides the fuel type could be important?

This is a valid point and the statement has been clarified by adding "and meteorology (i.e. temperature and wind speed)." to this sentence and the reference: Andrea 2019 (Lines 458-460 revised manuscript).

R2-C9 .p.20 ll.628-640: I suggest that you give your best guess

We are sorry, but we do not understand to what we should be giving our best guess. This part of the conclusions is outlining concerns with some of the assumptions used in biomass burning emissions estimations and stating that all uncertainties should be

fully described (as we have done by complete error propagation).

Language R2-C10. p.1 l.12: please correct: 'the contaminant to the atmosphere'

Changed to: "Mercury (Hg) emissions from biomass burning are an important Hg source to the atmosphere and an integral component of the global Hg biogeochemical cycle."

R2-C11. l.13: I suggest 'part' in stead of 'compartment'

The word used is component. We have left it as it was.

R2-C12. l.17: 'emissions plume' sounds odd, I suggest just 'plume'

Corrected.

R2-C13. l.19: '. . .,which is 2.4 times background concentration'

Corrected.

R2-C14. p.14 ll.424-430: Text formatting seems off here.

We do not see any formatting issues within this text.

Anonymous Referee #3

R3-C1. The authors present aircraft measurements of GEM, CO, CO2, CH4, and NMHCs in the plume of a forest fire in Saskatchewan. From these data they derive the emission ratios and calculate GEM emissions by three different methods. In addition, they calculate GEM flux using the screen flown downwind of the fire. The results of GEM emissions calculated by different methods are compared and their uncertainties discussed. The paper is well written but for publication it needs to provide more information at times, mentioned below. Some questions, listed below should also be answered:

We thank the reviewer for their positive description of the paper. Please note that after discussion between the co-authors the term NMHCs was updated to NMOGs (nonmethane organic gases) as the measurements also include oxidised organic gases; and hence this is the more correct terminology.

R3-C2. The purpose of the extensive discussion of the GEM enhancements is not clear. In addition, it will strongly depend on the meteorology which is omitted from the discussion. Without consideration of the windspeed (dilution) and the distance to the fire, the comparison with measurements published by other authors does not make much sense.

We acknowledge the reviewer's concern and have updated this paragraph. This includes removal of overly elaborative text and clarification of the issues highlighted by the reviewer. Lines 391-405 (revised manuscript) now states:

"The two other studies examining GEM concentrations in near-source, wildfire plumes using aircraft measured enhancements of ≈1.4 (Friedli et al., 2003a) and ≈6 (Friedli et al., 2003b) times background, placing the maximum enhancement observed in our study in the middle of those values. The size of the fires is likely to have played an important role in the differing enhancements, and indeed the burned area of fires were 1.7 km2 and 220 km2, respectively (compared to 88.0 km2 for the GLP fires). Additionally, both previous studies appear to have sampled the emissions plumes closer than Screen 1 of our flight. The differing distance of measurements from the fire (dilution effect) is another major factor driving the different enhancements between these fires. Other factors that are likely to affect the magnitude of GEM enhancement include extent of area burning during the sampling period, intensity (flaming or smouldering; potential change in PBM fraction) of the fire during the monitoring period, and/or variability in the concentration of Hg in the biomass of the different tree species being burned. Measurements collected from the ground-based Cape Point monitoring station in South Africa are the only other near source measurements reported from a wildfire emissions plume (23 km NNW of the site). This fire burned a very similar area to the GLP fires (≈ 90 km2) and GEM enhancements were ≈1.45x background (Brunke et al., 2001)."

[Figure]

Nonetheless, we believe completely omitting such a discussion on enhancements of near-source GEM measurements made in research aircraft would be eliminating comparison with the most relevant data available in the literature. Thus, we prefer to keep the discussion in its now revised form.

R3-C3. Section 2.1: Measurement of wind speed and direction onboard aircraft is not easy. The reader would like to know how these parameters were measured and with which uncertainties. This information is needed to assess the uncertainty of fluxes calculated by TERRA.

Gordon et al. (2015) describes the instrumentation that was used and section 4.1.1 gives a breakdown for estimating uncertainty. They did a Monte Carlo simulation varying the wind measurements that led to an uncertainty of about 1% on the final emission numbers related to variations in the wind measurements and the concentration of the compound. This ends up essentially being negligible when you consider the other uncertainties related to extrapolation below the flight path, but has been included in the calculation of the uncertainties of the TERRA estimates. Wind directions through Screen 1 (screen used for TERRA calculations) were relatively consistent. The following was added to line 155 of the revised manuscript: "...with a Rosemount 858 probe (see Gordon et al., 2015, for details)..."

Furthermore, the paragraph (lines 332-353 revised manuscript) was updated to improve the clarity of the discussion on TERRA uncertainties as suggested by the reviewer. This includes listing the 0.4 m s-1 uncertainty of the 32Hz wind speed measurements:

"In this study, we apply TERRA to the stacked horizontal legs of the flight track on the first screen downwind of the fire. Concentrations of Hg are extrapolated below the lowest flight altitude using a linear least-squares fit (recommended for ground-based emissions; Gordon et al., 2015) at each horizontal grid square below the lowest flight track in the plume area. Extrapolation below the flight path has been shown to be

the main source of uncertainty in TERRA. Two alternate extrapolation methods were tested: (i) including assuming a well-mixed layer (constant concentration) below the flight path and (ii) assuming a background concentration at the surface and linearly decreasing concentrations between the lowest flight track and the surface. There was less than 5 % difference in the resulting emission rates between these three methods of extrapolating data to the surface (we very conservatively estimate the extrapolation uncertainty to be 10 %).

The highest transect for Screen 1 downwind of the fire shows a consistent GEM background concentration along the whole transect. The consistent background of this highest transect indicates it was above the plume. Hence, there are no significant emissions above this height. The GEM concentrations measured during the spiral flown to determine the mixed layer height confirms this.

Although the uncertainty of 32 Hz wind speed measurements are ≈0.4 m s-1, when synchronised to lower frequency (1 Hz) mixing ratio measurements this uncertainty contributes <1 % to the overall uncertainty of the emissions transfer rate (Gordon et al., 2015) and likely less at the 2-min GEM data resolution. The overall emissions transfer uncertainty was conservatively estimated to be 15 % (4 % measured uncertainty from average GEM concentration from Screen 1; 1 % wind speed and between transect concentration interpolations; 10 % concentration extrapolation below screen). More details of the uncertainty estimations for TERRA are contained in Gordon et al. (2015) and Liggio et al. (2016)."

R3-C4. Sections 2.2 and 2.3: What are the estimated uncertainties of the individual GEM, CO, CO2, and CH4 measurements. These uncertainties are needed for assessment of the quality of ERs: e.g. the poorer quality of GEM/CH4 ratio could be caused by higher uncertainty of CH4 measurements? They are also needed for the orthogonal correlations (Cantrell, ACP, 8, 5477-5487, 2008).

The following two statements were added to the revised manuscript (Lines 207-209

and Lines 233-236) to clarify the instrument uncertainties:

1. "Uncertainty of this system was determined to be 3x the standard deviation ($3\sigma$) of the measurements made in background air (0.054 ng m-3; n = 30)."

2. "The $3\sigma$ values for CO, CO2, CH4, and NMOGs are 12, 380, 4, and 60 ppb, respectively, and were calculated using the same approach described for the Tekran 2537X. The instrument uncertainties are similar to those described and are outlined in more detail elsewhere (Gordon et al., 2015; Baray et al., 2018; Liggio et al., 2019; Karion et al., 2013).

The uncertainty of the CH4 measurements are the lowest of all instruments used; hence, they are not the source of the higher GEM:CH4 uncertainty. This uncertainty is most likely associated with the much lower proportional enhancements of CH4 compared to CO, NMOGs, and GEM. A sentence has been added to the revised manuscript to highlight this (lines 472-473): "Similar to CO2, CH4 is proportionally enhanced in the fire much less than GEM, CO, or NMOGs."

R3-C5. Line 233: I presume the GEM background is given as an average of 2 min measurements. What was the number of the GEM measurements used in this average?

"(. . .; n = 30)" has been added to the end of line 209 of the revised manuscript.

R3-C6. Lines 237-241: The consideration of only GEM enhancements >125% is probably not justified for several reasons: a) It is arbitrary – why not 115%? b) The selection of >125% GEM data may show only a part of the plume which may not be representative of the whole plume. c) With increasing distance to the fire, the section of plume with >125% would decrease relatively to the whole plume which again poses the question of representativity. d) The authors state that the data below 125% the enhancement are "too variable and too uncertain" to be considered. The concern about the uncertainty should not be the problem if the authors used orthogonal regression with uncertainties

of both GEM and X. The variability should also be no problem: the more points the smaller R is significant. Reference to Yokelson et al. (2013) is not quite appropriate for the situation here, i.e. with measurements up to 100 km distance from the fire with nearly constant background mixing ratios. In addition, the limitation to GEM enhancements >125% even seems to be unnecessary: the ERs in figure S4.1 calculated from all data and in Table 1 calculated with only GEM enhancements >125% are probably the same in statistical terms, i.e. cannot be distinguished taking into account the ERs uncertainties and the number of measurements. An additional table of ERs from Table 1 and ERs from figure S4.1 could be used to illustrate the necessity of the >125% threshold or its absence.

We refer the reviewer to the response to reviewer 1 comment 7 (R1-C9) above. We added a sensitivity assessment to our justification for using this value (Section S5). We argue that the Yokelson et al. (2013) study is appropriate because plume dilution occurs not only along the plume path, but also perpendicularly to this; the plume gets wider further from source. Again, we stress the reason for not using all data is that the co-contaminants are often at or below background and are not enhanced. This is demonstrated in the sensitivity analysis now included in Section S5. Additionally, the reviewer must consider that the majority of measurements <1.25x background are not inside any part of the plume. Many are measurements made during turns, traverses, or spirals between screens. Such data points are not spatially related to the fire plume where we assess emissions and concentrations from that plume.

Please note the term ">125%" was changed to "1.25x" throughout the text due to a question from another reviewer.

R3-C7. Paragraph lines 245-248: What type of regression was used: LSQF or orthogonal one? The usual LSQF should not be used to calculate ERs!!!

We thank the reviewer for identifying this very important issue. All regressions have been updated to follow the method developed by Neri et al. (1989) and the uncertainties of the regression slopes was determined by Reed (1989). This had minimal impact of the data (particularly for the data based on the GEM:CO and GEM:CO2 ERs). Nonetheless, it is the correct way to make the calculations and all data and figures that used these ERs have been updated throughout the text and the supplementary information.

R3-C8. The last sentence is difficult to understand because any type of regression automatically adjusts for the backgrounds – it is just a shift in the coordinate system.

We totally agree, it is just a shift in the coordinate system. However, for clarity we prefer to leave the sentence.

R3-C9. Probably CO, CO2, CH4, and NMHC measurements were converted to 2 min averages synchronized with 2 min GEM for correlations? If so, it should be mentioned.

This is mentioned (lines 230-233 revised manuscript); however, the words "...synchronised and..." were added to this sentence for clarity.

R3-C10. Paragraph lines 249-252: How was the integration made? Only for the enhancements >125%?

Yes, the integrations used the same >1.25x GEM background data set. This is stated in the previous paragraph. The method used was that described in Urbanski 2013 (and stated in the text). As this method was only used for comparative purposes (to ensure estimates using the two methods were not substantially different), we do not believe it necessary to go into further details describing this method that is outlined elsewhere.

R3-C11. Paragraph lines 305-327: More information is needed in the description of TERRA calculation? How were GEM measurements (2 min) interpolated? The treatment of the layer above the highest transection and the inversion layer has to be mentioned too. Some of this information is provided in Section 3.4 but the reader would expect it here.

The GEM concentrations were interpolated using a simple Kriging method as already

described in the manuscript. The following additions were made to the TERRA method description in Section 2.6:

Lines 328-331 in revised manuscript: "Pollutant and wind data are mapped to a virtual screen (only Screen 1 of flight) and concentration data interpolated using a simple kriging function. For the time series input into TERRA, the 2-min and 2-sec data becomes 1 second data; each second during these 2-min or 2-sec periods has the same concentration."

(As mentioned in comment R3-C3) Lines 332-353 in the revised manuscript were updated to better described the TERRA uncertainties including treatment of concentrations above the highest transect (highest transect showed background concentrations).

R3-C12. Lines 348 and 349: How was the correlation made: orthogonal? 2min data? What are the regression lines?

Please see responses R3-C4 and R3-C7.

R3-C13. Paragraph lines 459-464: The conversion of 2 min GEM measurements into 0.5 Hz data using the GEM/CO ratio was probably made mainly for the TERRA calculations. If so, it should be mentioned.

The conversion of GEM to 0.5 Hz resolution was made for a number of reasons. (i) TERRA modelling (although only used for comparison – 2-min GEM data was used for the final estimate), (ii) to estimate the GEM concentrations at a higher temporal resolution (0.5 Hz) based on the robust GEM:CO ER, and (iii) to estimate concentrations at the fire front. The manuscript states the intent when each instance is used.

R3-C14. Paragraph 465-480: The text here is highly speculative because the estimations are strongly dependent on the meteorology which is not mentioned. In addition, it does not contribute much to the purpose of the paper.'

This meteorology uncertainty (stated: variable wind speed) and other uncertainties (stated: extrapolation, distance uncertainty) are stated in the last sentence of this

paragraph. While we acknowledge these sources of uncertainty (as we do in the manuscript), we believe it does add to the manuscript as it is a direct benefit of the experimental flight design of flying a number of screens downwind of the plume. Other studies that have performed more random flight designs that do not track air-parcels cannot make such estimates. As such, our preference is to keep this discussion in the study.

R3-C15. Table 1: The comparison of maximal measured GEM enhancements is strongly dependent on their temporal resolution, as shown in this work, and on the meteorological parameters (especially wind speed in combination with the distance to the fire, i.e. dilution). Without taking all these factors into account, the comparison does not make much sense and as such should be deleted from the table, and also from the text.

The reviewer is assuming wind speed/dilution are the only factors influencing the GEM enhancements of these fires. We have described this in the manuscript and that there are other contributing factors as to why the enhancements are potentially different. While we acknowledge this is definitely not the most impactful data of the study, we suggest that there are readers who will find this data useful and use it in calculations, modelling or future studies on forest fire emissions. Our preference is that the data remain, particularly with the amendments made according to the reviewer's comment R3-C2.

R3-C16. Table 2: "Uncertainty" should be used instead of "error" here and throughout the text. The calculation of uncertainties and the terms used in the equation 7 should be given either in the manuscript or in the supporting information.

This has been corrected in the table and throughout the manuscript.

R3-C17. Fig. 2b: I wonder about 2s GEM data derived from 0.5 Hz CO data using the GEM/CO ratio. What is this conversion good for? I find its presentation misleading because it pretends much higher density of GEM data than available. The 0.5 Hz CO,

CO2, and CH4 measurements should be converted to the 2 min GEM time stamp, at least for the regressions.

We refer the reviewer to comment R3-C13. We have a species measured at higher temporal resolution (CO). The GEM:CO emissions ratio based on the 2-min resolution data has the lowest uncertainty of the measured co-pollutants (and from out review of the literature – in any published study). We feel that this is justifiable and effective manner to express GEM concentrations at this higher temporal resolution. We do not feel we are misleading the readers as we have explained the rationale in the text and in the Fig 2b caption in which we state this is the 2-second GEM concentration calculated by conversion of the 0.5 Hz CO data using the GEM:CO emissions ratio. This is important because it shows the reader that concentrations at higher temporal resolution give higher values – rather than the "time-averaged" 2-min samples of the instrument. This may be obvious to the reviewer, but it may be less clear to some readers.

R3-C18. Fig. 3: Which type of regression was used? The usual one (LSQF) or one which considers the X and the Y uncertainties? What was the number of correlated points: R without the number of points does not say anything about the significance of the regression. Ditto the figure S4.1.

Orthogonal or total least squares regressions have now been applied using the instrument uncertainties. The number of data points used is given in the methods section. Data >1.25x is numbered: 24. When all data are considered n = 120. Both these n-values have been added to the captions of the respective figures.

R3-C19. Fig. 5: "Distance" in the name of x axis may be easily mixed up with the distance to fire.

It is the estimated distance [of the measurement] from [the] fires and is stated as such. Furthermore, the figure caption states:

"The maximum 2-sec calculated GEM concentration derived from GEM:CO ER for each screen and the estimated distance this measurement was from the GLP fires."

We hope this clarifies the reviewers concern.

  REFERENCES:

Baray, S., Darlington, A., Gordon, M., Hayden, K. L., Leithead, A., Li, S. M., ... & McLaren, R.: Quantification of methane sources in the Athabasca Oil Sands Region of Alberta by aircraft mass balance. Atmospheric Chemistry and Physics, 18, 7361-7378, 2018.

Giglio, L., Randerson, J. T., and van der Werf, G. R.: Analysis of daily, monthly, and annual burned area using the fourth‐generation global fire emissions database (GFED4). J. Geophys. Res. Biogeosci., 118, 317-328, https://doi.org/10.1002/jgrg.20042, 2013.

Gordon, M., Li, S.-M., Staebler, R., Darlington, A., Hayden, K., O'Brien, J. and Wolde, M.: Determining air pollutant emission rates based on mass balance using airborne measurement data over the Alberta oil sands operations. Atmos. Meas. Tech., 8, 3745–3765, https://doi.org/10.5194/amt-8-3745-2015, 2015.

Karion, A., Sweeney, C., Wolter, S., Newberger, T., Chen, H., Andrews, A., ... & Tans, P.: Long-term greenhouse gas measurements from aircraft. Atmospheric Measurement Techniques, 6(3), 511-526, 2013.

Liggio, J., Li, S. M., Staebler, R. M., Hayden, K., Darlington, A., Mittermeier, R. L., ... & Vogel, F.: Measured Canadian oil sands CO 2 emissions are higher than estimates made using internationally recommended methods. Nature communications, 10(1), 1-9, 2019.